# C-3PO: Compact Plug-and-Play Proxy Optimization to Achieve Human-like Retrieval-Augmented Generation

Guoxin Chen [1 2]   Minpeng Liao [2 *]   Peiying Yu [3]   Dingming Wang [4]
Zile Qiao [2]   Chao Yang [5]   Wayne Xin Zhao [1 *]   Kai Fan [2 *]

## Abstract

Retrieval-augmented generation (RAG) systems face a fundamental challenge in aligning independently developed retrievers and large language models (LLMs). Existing approaches typically involve modifying either component or introducing simple intermediate modules, resulting in practical limitations and sub-optimal performance. Inspired by human search behavior—typically involving a back-and-forth process of proposing search queries and reviewing documents, we propose C-3PO, a proxy-centric framework that facilitates communication between retrievers and LLMs through a lightweight multi-agent system. Our framework implements three specialized agents that collaboratively optimize the entire RAG pipeline without altering the retriever and LLMs. These agents work together to assess the need for retrieval, generate effective queries, and select information suitable for the LLMs. To enable effective multi-agent coordination, we develop a tree-structured rollout approach for reward credit assignment in reinforcement learning. Extensive experiments in both in-domain and out-of-distribution scenarios demonstrate that C-3PO significantly enhances RAG performance while maintaining plug-and-play flexibility and superior generalization capabilities. Code is available at https://github.com/Chen-GX/C-3PO.

## 1. Introduction

Recent advances in retrieval-augmented generation (RAG) for large language models (LLMs) have demonstrated remarkable capabilities in various tasks (Anthropic, 2024; Hurst et al., 2024; Dubey et al., 2024; Yang et al., 2024a; Mesnard et al., 2024; Asai et al., 2024; Chen et al., 2024a;b; Wei et al., 2025; Sun et al., 2025b;a), empowering LLMs to acquire up-to-date or domain-specific knowledge while mitigating hallucinations (Gao et al., 2023; Fan et al., 2024; Qiao et al., 2024). The effectiveness of RAG systems, however, hinges on the alignment[1] between the retriever and the LLM—an inherently challenging goal as these components are typically developed independently without co-training. This lack of co-training can result in semantic mismatch and suboptimal interactions: retrievers may fail to provide information tailored to the LLM's needs, while LLMs may struggle to generate effective queries or seamlessly incorporate retrieved content.

Existing approaches address this misalignment through three main strategies: (1) fine-tuning retrievers to align with LLM preferences, (2) optimizing LLMs to adapt to retriever behavior, and (3) introducing intermediate modules to bridge the gap between them (Ma et al., 2023; Shi et al., 2024; Asai et al., 2024; Wei et al., 2025; Yu et al., 2024a;b). Despite progress, these methods face notable challenges: fine-tuning retrievers often requires carefully curated data and may not be feasible for commercial search engines (Schmidt, 2014; Nakano et al., 2021), while optimizing LLMs is resource-intensive and risks compromising their original capabilities (Zhou et al., 2024). Approaches that introduce intermediate modules to avoid modifying either the retriever or the LLM primarily focus on optimizing individual tasks, such as query rewriting or document reranking (Ma et al., 2023; Wang et al., 2023; Tan et al., 2024). However, optimizing a single task in isolation often leads to suboptimal results, as the effectiveness of RAG systems relies on the cohesive interaction and collaboration among multiple components throughout the entire pipeline (Fan et al., 2024; Zhou et al., 2024).

In human's search behavior, the process typically involves an iterative back-and-forth process of proposing search queries and reviewing documents until the correct response is either found in the retrieved documents or emerges in the person's mind. Similarly, LLMs can emulate this process

---

*Corresponding Authors. [1]Gaoling School of Artificial Intelligence, Renmin University of China [2]Tongyi Lab [3]Soochow University [4]University of Oxford [5]Tsinghua University. Correspondence to: Guoxin Chen <gx.chen.chn@gmail.com>, Minpeng Liao <minpeng.lmp@alibaba-inc.com>, Wayne Xin Zhao <batmanfly@gmail.com>, Kai Fan <k.fan@alibaba-inc.com>.

*Proceedings of the 42nd International Conference on Machine Learning*, Vancouver, Canada. PMLR 267, 2025. Copyright 2025 by the author(s).

[1]Here, alignment refers to the functional coordination between retrievers and LLMs, rather than human preference alignment.

by taking on multiple roles within a search pipeline: proposing search queries, reviewing documents, deciding when to terminate retrieval, and generating the final response, among other tasks. However, assigning all these tasks to LLMs results in numerous calls, leading to high computational costs, especially for complex questions. To address this, it is desirable to design a compact proxy capable of handling most tasks, while reserving the most challenging ones to LLMs—such as planning the overall roadmap and generating the final response.

Therefore, we propose C-3PO, a proxy-centric alignment framework that employs a lightweight yet effective proxy to facilitate seamless communication between retrievers and LLMs without modifying them or compromising their original capabilities. As illustrated in Figure 1, C-3PO integrates a lightweight multi-agent collaborative system within a single proxy model, where multiple agents work in a human-like manner to assist the entire RAG pipeline. To optimize this proxy, we employ multi-agent reinforcement learning (MARL) for end-to-end training, treating the retrievers and LLMs as part of the environment. To address the key challenge of optimizing multiple agents with distinct tasks, we introduce a tree-structured rollout mechanism and Monte Carlo credit assignment to improve reward distribution among different agents. In this way, our C-3PO redistributes the sampling efforts from the question level to the action level, enabling more efficient credit assignment in multi-agent systems through expectation-based reward distribution. Experimental results demonstrate that the RL-trained proxy achieves robust performance on both in-domain and out-of-distribution datasets, even with unseen retrievers and LLMs, highlighting its plug-and-play modularity and superior generalization capability. Our contributions are summarized as follows:

- We propose C-3PO, a novel proxy-centric alignment framework that bridges the gap between retrievers and LLMs through a lightweight multi-agent system, enabling seamless integration without modifying existing RAG components.

- We design an efficient multi-agent collaborative system within a single proxy model that emulates human-like search behavior, where specialized agents handle different aspects of the RAG pipeline while maintaining computational efficiency.

- We develop an innovative training approach combining MARL with a tree-structured rollout mechanism and Monte Carlo credit assignment, effectively addressing the challenge of optimizing multiple agents towards the system-level performance in an end-to-end manner.

- Extensive experiments demonstrate that C-3PO achieves superior performance across diverse datasets and exhibits strong generalization capability with unseen retrievers

and LLMs, validating its effectiveness as a plug-and-play solution for RAG systems.

## 2. Related Work

**Retrieval-Augmented Generation.** Retrieval-augmented generation (RAG) has emerged as a crucial technique for enhancing LLMs' capabilities by incorporating external knowledge sources (Gao et al., 2023; Yu et al., 2024b). Recent studies have highlighted the importance of aligning the retriever with the LLM to achieve superior performance (Fan et al., 2024; Chan et al., 2024). This alignment can be approached through three main strategies: retriever fine-tuning methods (Shi et al., 2024), LLM fine-tuning methods (Asai et al., 2024; Wei et al., 2025; Yu et al., 2024a), and intermediate modules methods (Ma et al., 2023; Wang et al., 2023; Tan et al., 2024). However, these methods often face practical limitations, such as focusing on local optimization, the inability to align with commercial search engines, and the substantial computational costs of LLM optimization. Different from previous work, we introduce a lightweight, proxy-centric alignment framework that implements alignment without modifying either the retriever or LLM while optimizing the entire RAG pipeline holistically.

**Multi-agent Systems.** Multi-agent systems have recently garnered increasing attention, especially in the context of complex task-solving and decision-making (Guo et al., 2024; Zhang et al., 2024; Chen et al., 2025). A major challenge in multi-agent frameworks is credit assignment—determining each agent's contribution to the overall system performance—which becomes particularly crucial in multi-agent reinforcement learning (Yuan et al., 2023a; Zhu et al., 2024). In our work, we propose Monte Carlo credit assignment mechanism to distribute system-level rewards to each agent in the form of expectations, enabling effective coordination within agents.

## 3. Preliminaries

Before introducing C-3PO, we first review the preliminaries of cooperative multi-agent reinforcement learning (MARL), where multiple agents work collaboratively to accomplish given tasks. A cooperative MARL problem is generally formalized as a cooperative stochastic game, represented by a tuple $\langle \mathcal{N}, \{\mathcal{S}^i\}_{i\in\mathcal{N}}, \{\mathcal{A}^i\}_{i\in\mathcal{N}}, \mathcal{T}, \mathcal{R} \rangle$, where:

- $\mathcal{N} = \{1, 2, ..., n\}$ is the set of agents in the system.

- $\mathcal{S}^i$ is the state space of agent $i$, where each agent maintains its agent-specific state information.

- $\mathcal{A}^i$ is the action space of agent $i$, defining the joint action space $\mathcal{A} := \mathcal{A}^1 \times \cdots \times \mathcal{A}^n$.

- $\mathcal{T} : \{\mathcal{S}^i\}_{i\in\mathcal{N}} \times \mathcal{A} \to \{\mathcal{S}^i\}_{i\in\mathcal{N}}$ is the deterministic state transition function, specifying how the states of agents

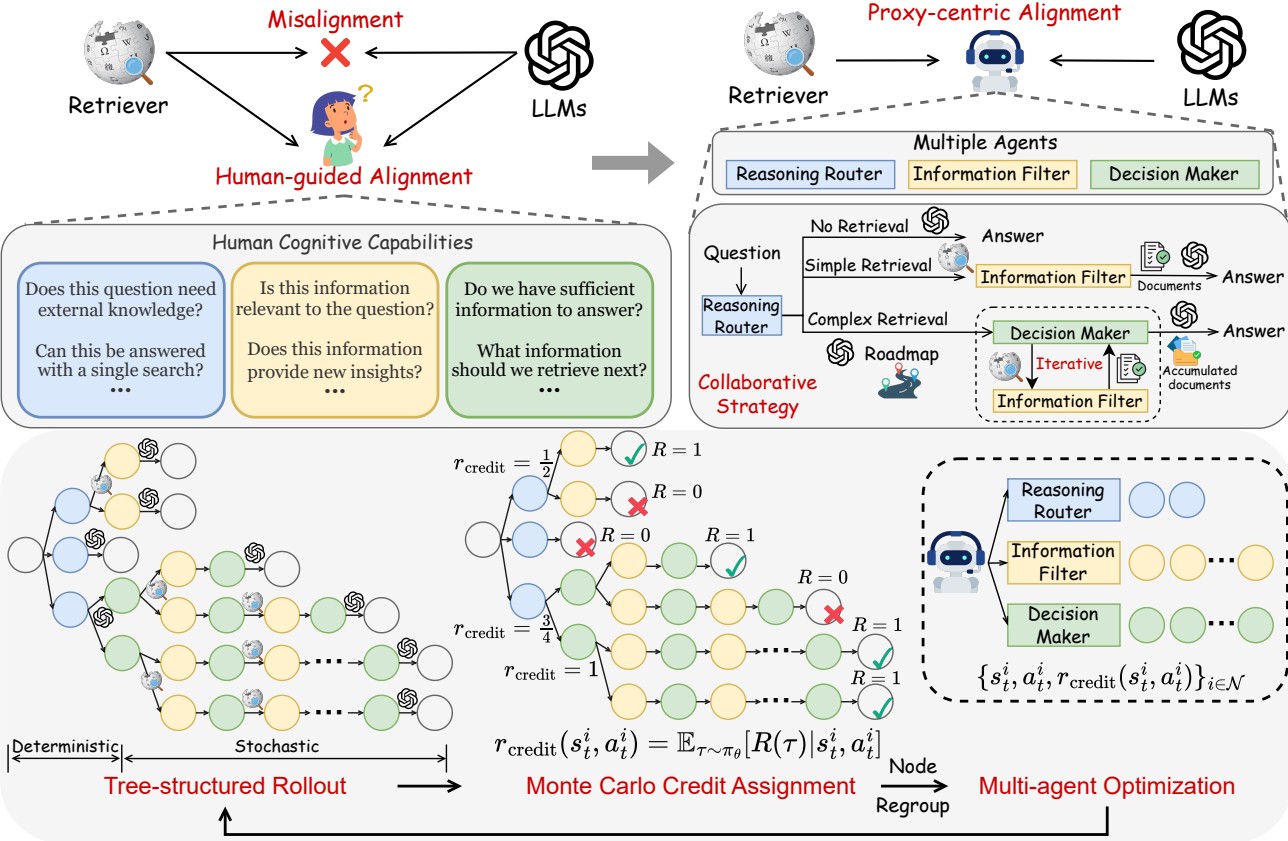

Figure 1. Overall framework of C-3PO. (Upper left) Essential cognitive capabilities required for effective RAG system interaction in human-guided alignment. (Upper right) Our proxy-centric alignment simulates these human-like interaction through a lightweight multi-agent system with collaborative strategies. (Bottom) The end-to-end optimization pipeline for our multi-agent system.

update given their joint action $\boldsymbol{a} \in \mathcal{A}$.

- $\mathcal{R} : \{\mathcal{S}^i\}_{i \in \mathcal{N}} \times \mathcal{A} \times \mathcal{N} \to \mathbb{R}$ is the system-level reward that measures the overall task completion, where 1 indicates success and 0 otherwise.

In this cooperative setting, all agents share the same system-level reward $\mathcal{R}$ and work together to accomplish the task. Each agent follows its policy $\pi^i : \mathcal{S}^i \to \mathcal{A}^i$ to select actions based on its local observations. A trajectory $(\{s_0^i\}_{i \in \mathcal{N}}, \boldsymbol{a}_0, \{s_1^i\}_{i \in \mathcal{N}}, \boldsymbol{a}_1, ...)$ represents the sequence of agent states and joint actions, where $s_t^i \in \mathcal{S}^i$ is the state of agent $i$ at time step $t$, and $\boldsymbol{a}_t = \{a_t^i\}_{i \in \mathcal{N}} \in \mathcal{A}$ is the joint action at time step $t$ under the joint policy $\pi = (\pi^1, ..., \pi^n)$.

## 4. Cooperative Multi-agent System

In this section, we elaborate on the role of each agent (Section 4.1) with their collaborative strategies (Section 4.2).

### 4.1. Multiple Agents

Inspired by human behavior mentioned in the Introduction, we design three specialized agents—Reasoning Router, Information Filter, and Decision Maker—to facilitate commu-

nication between the retriever and the LLM, as illustrated in Figure 1. These agents operate as distinct roles within a single lightweight proxy using targeted instructions, collaboratively managing various aspects of the RAG pipeline. This unified design ensures efficient coordination while maintaining simplicity for edge deployment. Formally, we define each agent as follows:

This proxy plays the role of **Reasoning Router** agent to determine the optimal reasoning strategy for a given question from a high-level perspective. Given the current state (the question), it selects actions using a maximum two-step operation:

1. **Decide Retrieval Necessity**: If the agent outputs [No Retrieval], the question is directly processed by the LLM, leveraging its internalized knowledge.

2. **Determine Question Complexity**: If the agent outputs [Retrieval], the agent also evaluates whether the question requires complex reasoning.

*For simple questions*, the agent continues to generate a single <query content> and interacts with the retriever to obtain documents. The retrieved documents are then processed by the **Information Filter** agent to extract relevant,

LLM-friendly content. Finally, the LLM uses the filtered documents to generate an answer to the question.

*For complex questions*, the agent outputs `[Planning]`, which will trigger a multi-step reasoning strategy that requires coordination with multiple agents. Further details on this strategy will be introduced later.

This proxy plays the role of **Information Filter** agent to process and filter retrieved information for identifying content suitable for LLMs. Its state space includes the question, the retrieved documents, and the current reasoning objective (if operating in `[Planning]` mode). Based on the given state, the agent selects an action to analyze and filter relevant documents using the following structured format:

> Thought: <Analysis of each documents>
> Action: [<Selected document IDs>]

This proxy plays the role of **Decision Maker** agent to determine the optimal action within the `[Planning]` strategy based on the current state. Its state space includes the question, the LLM-generated roadmap, and the accumulated documents from the reasoning history. Given the current state, the agent selects an action to evaluate progress and decide the next operation, using the following structured format:

> Thought: <Analysis of current progress and objective>
> Action: {[Retrieval]<subquery content> (Continue with retrieval-filter loop), or [LLM] (Pass to LLM for answering)}

### 4.2. Collaborative Strategy

With the three specialized agents defined, we now describe how they collaborate to efficiently handle different types of questions. Their coordination follows a structured workflow, enabling adaptive and multi-granular information processing through various strategies, as detailed below.

The **Direct Answering Strategy** and **Single-pass Strategy** have already been introduced in the definition of the **Reasoning Router** agent, corresponding to the agent outputs `[No Retrieval]` and `[Retrieval]<query content>`, respectively.

**Multi-Step Reasoning Strategy** corresponds to the `[Planning]` output by **Reasoning Router** agent. Designed to address complex questions requiring a high-level roadmap from LLM and multiple retrieval-filter loops, this strategy enables iterative information gathering and reasoning through the following three phases:

1. **Generate Roadmap**: The LLM decomposes the complex question into a structured set of subgoals, providing high-level guidance for the proxy.

2. **Iterative Retrieval-filter Loop**: Guided by the roadmap, the **Decision Maker** evaluates the current progress, determines the next objective, and generates subqueries for the retrieval-filter loop. This process is carried out in coordination with the **Information Filter** and continues iteratively until the **Decision Maker** determines that the accumulated documents contain sufficient information to address all subgoals.

3. **Final Answer**: All accumulated information is passed to LLM for answer generation.

Notably, generating the roadmap is the only role that is not played by the proxy in our communication pipeline; however, the LLM is invoked only once in the `[Planning]` strategy, minimizing computational overhead. Additionally, it is important to note that the number of retrieval-filter loops may not directly correspond to the number of subgoals, as a single retrieval might address multiple subgoals or require multiple attempts for a single subgoal.

Through these three strategies, our multi-agent system adaptively handles questions of varying complexity. The Reasoning Router automatically selects the most appropriate strategy based on the characteristics of each question: the Direct Answering Strategy provides immediate responses for general knowledge, the Single-pass Strategy efficiently retrieves information for fact-based questions, and the Multi-Step Strategy addresses complex questions through guided iterative reasoning. This hierarchical approach ensures optimal resource utilization by aligning computational effort with question complexity while potentially maintaining high response quality. Next, the key focus is to optimize the proxy to learn the knowledge boundaries of the LLM and master the specific capabilities of each agent.

## 5. Multi-Agent Proxy Optimization

Since the final answer generated by the LLM is straightforward to evaluate as the system-level reward, it is intuitive to employ reinforcement learning to optimize the proxy. However, each agent within the proxy serves as an intermediate module, responsible for only a partial trajectory of the RAG pipeline. This makes it difficult to define agent-level rewards (Guinet et al., 2024). For example, a high-quality generated query might still result in a low system-level reward due to poor subsequent document filtering. To tackle this challenge, we propose a tree-structured rollout approach for robust on-policy learning, utilizing deterministic rollout in the early stages and stochastic rollout in later stages.

### 5.1. Credit Assignment

To avoid the sparse reward in traditional single trajectory rollout, we propose the tree-structured rollout for credit

assignment, which distributes system-level rewards across agents to mitigate the high variance of local rewards for each agent. Through this way, C-3PO effectively redistributes the sampling effort from the question level to the action level while maintaining a similar computational budget. The core idea is to evaluate each agent's contribution by forcing the Reasoning Router to explore all possible reasoning strategies during the rollout for each question, and enable information sharing across different reasoning paths through action-level exploration.

**Deterministic Rollout.** Given a question $q$, we begin by forcing the Reasoning Router to explore both [No Retrieval] and [Retrieval]. For the [Retrieval] branch, we further force the agent to explore simple reasoning by directly generating <query content> and complex reasoning through [Planning]. As shown in Figure 1, we deterministically construct a decision tree during the first stage of the rollout. Current tree, with a depth of 2, enables simultaneous exploration of multiple reasoning paths, providing a comprehensive understanding of how each decision impacts the final outcome.

**Stochastic Rollout.** Once the overall reasoning strategy is confirmed, the subsequent rollout employs sampling to expand the decision tree. For each non-terminal node, we randomly sample $K(t)$ candidate actions from the proxy $\pi$ for the $i$-th agent at depth $t$.[2] Specifically, the tree is expanded using the following children (actions):

$$\{a_{t,k}^i\}_{k=1}^{K(t)} \sim \pi(\cdot|s_t^i, \text{instruction}_i) \quad (1)$$

$$K(t) = \begin{cases} 2, & \text{if } t \le 4 \\ 1, & \text{if } t > 4 \end{cases} \quad (2)$$

where instruction$_i$ refers to the task-specific instruction for the $i$-th agent, and $K(t)$ represents the dynamic branching factor at depth $t$, balancing exploration and computational efficiency. Each sampled action $a_{t,k}^i$ triggers a state transition governed by:

$$s_{t+1,k}^j = \mathcal{T}(s_t^i, a_{t,k}^i), \quad (3)$$

where $j \in \mathcal{N}$ denotes the next agent in the predefined strategy (Section 4.2).[3] By recursively applying this sampling process until leaf nodes are reached, we construct a decision tree containing multiple trajectories $\tau$:

$$\tau = \{(s_t^i, a_{t,k}^i, s_{t+1,k}^i)\}_{i \in \mathcal{N}, t, k} \quad (4)$$

where each leaf node includes the final system-level reward ($R = 1$ for success and $R = 0$ for failure).

---

[2]The time step $t$ is reused as the depth of the tree. Since nodes are expanded layer by layer in our implementation, the depth corresponds to the time step.

[3]The transition function $\mathcal{T}$ is deterministic and involves state updates. See Appendix B.2 for more details.

**Monte Carlo Credit Assignment.** Instead of constructing a single trajectory rollout for each question, we create a *tree-structured rollout* containing multiple trajectories. This structure enables us to trace how individual decisions impact the system-level outcome. For each agent-generated node $(s_t^i, a_t^i)$, we compute its credit reward based on the expected system-level reward:

$$r_{\text{credit}}(s_t^i, a_t^i) = \mathbb{E}_{\tau \sim \pi_\theta}[R(\tau)|s_t^i, a_t^i] \approx \frac{\sum_{l \in L(s_t^i, a_t^i)} R_l}{|L(s_t^i, a_t^i)|}, \quad (5)$$

where $L(s_t^i, a_t^i)$ denotes the set of leaf nodes reachable from $(s_t^i, a_t^i)$, and $R_l$ is the final reward of leaf node $l$.

Our proposed credit assignment mechanism offers several key advantages over a single trajectory rollout: (1) Our rollout thoroughly explores all possible strategies for each question, generating numerous intermediate training examples for each agent. (2) Most importantly, while directly redistributing a single system-level reward to agent nodes in a single trajectory is challenging, our approach accurately estimates the reward of each agent node in a probabilistic expectation using the tree-structured rollout.

### 5.2. Training Method

For each sampled tree, we disassemble it into individual nodes and group them by their corresponding agents, as illustrated in Figure 1. The token sequence within each node, along with its corresponding reward computed via Eq (5), is added to the replay buffer for Proximal Policy Optimization (PPO) (Schulman et al., 2017). The overall training objective follows the standard PPO framework but incorporates multi-agent aggregation.

$$\mathcal{L}_{\text{C-3PO}} = \sum_{i \in \mathcal{N}} \mathcal{L}_{\text{PPO}}^i, \quad (6)$$

where the loss $\mathcal{L}_{\text{PPO}}$ can represent either value loss or policy loss. Details of the loss functions are provided in the Appendix A.1. Following the PPO recipe in Ouyang et al. (2022), the PPO for LLMs typically requires an initialization from the supervised fine-tuning model. Therefore, we employ our tree-structured rollout with rejection sampling (Yuan et al., 2023b) to collect seed data and use cross-entropy loss for the supervised warm-up phase.

## 6. Experiments

### 6.1. Experimental Setup

**Datasets.** To comprehensively evaluate our C-3PO, we experiment on both single-hop datasets including Natural Questions (NQ) (Kwiatkowski et al., 2019), PopQA (Mallen et al., 2023), and TriviaQA (TQA) (Joshi et al., 2017), as well as multi-hop datasets including 2WikiMultiHopQA

*Table 1.* Main results. Training / Testing with Wiki Retriever.

| Methods | LLM Server | Proxy | Tuned Params | Multi-hop | | | Single-hop | | | Average |
|---|---|---|---|---|---|---|---|---|---|---|
| | | | | 2Wiki | HQA | Musique | NQ | PopQA | TQA | |
| Direct | Qwen2-72B | ✗ | - | 41.6 | 45.9 | 20.5 | 53.3 | 24.3 | 76.3 | 43.65 |
| Standard | Qwen2-72B | ✗ | - | 34.4 | 55.7 | 41.1 | 60.8 | 38.0 | 78.1 | 51.35 |
| *Retriever Fine-tuning* | | | | | | | | | | |
| REPLUG | Qwen2-72B | ✗ | 109M | 33.5 | 50.4 | 31.9 | 55.6 | 39.3 | 77.6 | 48.05 |
| *LLM Fine-tuning* | | | | | | | | | | |
| Self-RAG | Qwen2-7B | ✗ | 7B | - | - | - | 50.4 | 38.5 | 66.2 | 51.70 |
| InstructRAG | Qwen2-7B | ✗ | 7B | 35.8 | - | - | 48.1 | 39.5 | 66.6 | 47.50 |
| Auto-RAG | Qwen2-7B | ✗ | 7B | 41.8 | 37.8 | - | 53.4 | 36.8 | 63.3 | 46.62 |
| Self-RAG | Qwen2-72B | ✗ | 72B | - | | - | 56.6 | 40.3 | 78.2 | 58.33 |
| InstructRAG | Qwen2-72B | ✗ | 72B | 49.7 | - | - | 57.8 | 40.1 | 77.9 | 56.37 |
| Auto-RAG | Qwen2-72B | ✗ | 72B | 47.7 | 45.7 | - | 55.3 | 40.9 | 72.1 | 52.34 |
| *Intermediate Module* | | | | | | | | | | |
| Reranker | Qwen2-72B | Qwen2-7B | 7B | 32.8 | 46.4 | 22.6 | 57.2 | 20.9 | 76.3 | 42.70 |
| QueryRewrite | Qwen2-72B | Qwen2-1.5B | 1.5B | 36.2 | 55.8 | 40.2 | 62.3 | 36.7 | 78.9 | 51.68 |
| SKR-KNN | Qwen2-72B | ✗ | - | 40.6 | 55.6 | 41.3 | 61.9 | 38.8 | 78.1 | 52.71 |
| SlimPLM | Qwen2-72B | Qwen2-7B | 3×7B | - | - | 24.4 | 62.4 | - | 76.2 | 54.33 |
| C-3PO (Ours) | Qwen2-72B | Qwen2-0.5B | 0.5B | **66.1** | 66.8 | 49.4 | 63.7 | **46.1** | 80.4 | 62.08 |
| | | | | ↑16.4 | ↑11.0 | ↑8.1 | ↑1.3 | ↑5.2 | ↑1.5 | |
| C-3PO (Ours) | Qwen2-72B | Qwen2-1.5B | 1.5B | 65.2 | **69.0** | **54.2** | **65.9** | 44.8 | **82.1** | **63.53** |
| | | | | ↑15.5 | ↑13.2 | ↑12.9 | ↑3.5 | ↑3.9 | ↑3.2 | |

(2Wiki) (Ho et al., 2020), Musique (Trivedi et al., 2022), and HotpotQA (HQA) (Yang et al., 2018). For each dataset, we only use 6000 randomly sampled questions instead of the full training set.

**Baseline.** We compare our method against a diverse set of baselines, including (1) **Direct**: Directly answer questions without retrieval. (2) **Standard RAG**: The standard retrieval-augmented method retrieves documents based on the question. (3) **Retriever Fine-tuning Method**: RE-PLUG (Shi et al., 2024). (4) **LLM Fine-tuning Method**: Self-RAG (Asai et al., 2024), InstructRAG (Wei et al., 2025), and Auto-RAG (Yu et al., 2024a). (5) **Intermediate module Method**: Reranker (Li et al., 2023), QueryRewrite (Ma et al., 2023), SKR (Wang et al., 2023), and SlimPLM (Tan et al., 2024).

**Implementation Details.** Following Asai et al. (2024), we construct our retrieval system using the 2018 Wikipedia dump (Yang et al., 2018) as the knowledge source and use contriever-msmarco (Izacard et al., 2022) as our dense retriever. We utilize Qwen2-72B-Instruct (Yang et al., 2024a) as fixed LLM server, while Qwen2-0.5B or Qwen2-1.5B is trained as candidate lightweight proxy for efficient edge deployment. In the warm-up phase, we collect 4 solutions for each question with Qwen2-72B-Instruct. We use a learning rate of 4e-5, with 3 epochs and a batch size of 512. For the RL phase, we set learning rate of 5e-7 for policy model and 5e-6 for value model with a batch size of 1024 and maximal depth of 13. More details please refer to Appendix A.

### 6.2. Main Results

We report the performance on both single-hop and multi-hop datasets in Table 1. **First**, our C-3PO consistently outperforms various baselines across different datasets, achieving superior average performance of 62.08% and 63.53% with lightweight proxies of only 0.5B and 1.5B parameters, respectively. This demonstrates the effectiveness of our proxy-centric alignment approach in bridging the gap between the retriever and the LLM. **Second**, compared to single-hop datasets, our method yields particularly notable gains in challenging multi-hop reasoning tasks. Specifically, C-3PO achieves significant improvements on multi-hop datasets (2Wiki +15.5%, HQA +13.2%, Musique +12.9%), while maintaining strong performance on single-hop tasks (NQ +3.5%, PopQA +3.9%, TQA +3.2%). This significant performance gain suggests that, even without training original RAG system, our C-3PO effectively enhances the coordination between the retriever and LLM, which is particularly crucial for addressing complex multi-hop tasks. **Third**, although retriever fine-tuning method (Shi et al., 2024) requires fewer tuned parameters, it does not overcome the limitations of standard RAG systems in handling complex cognitive and multi-hop reasoning tasks. Both LLM fine-tuning and intermediate module methods show promising results, but are constrained by either large tuning parameters (7B/72B) or inconsistent performance across different reasoning datasets. In contrast, our C-3PO achieves consistent improvements across all datasets with only 0.5B/1.5B additional parameters, demonstrating both efficiency and effectiveness in enhancing RAG systems.

*Table 2.* Out-of-Distribution results. Testing with Google Search Engine.

| Dataset / LLM Servers | OOD Datasets + Retrieval | | OOD Datasets + Retrieval + LLM Server | | | | | | Average |
|---|---|---|---|---|---|---|---|---|---|
| | FQA | M-RAG | FQA | M-RAG | FQA | M-RAG | FQA | M-RAG | |
| | Qwen2-72B | | Qwen2-7B | | Llama3.3-70B | | GPT-4o-mini | | |
| Direct | 58.2 | 42.2 | 40.6 | 34.4 | 60.8 | 43.8 | 58.8 | 51.7 | 48.81 |
| Standard | 66.4 | 46.3 | 57.4 | 39.2 | 65.6 | 41.7 | 71.6 | 52.3 | 55.06 |
| *LLM Fine-tuning* | | | | | | | | | |
| Self-RAG | 49.4 | 39.2 | - | - | - | - | - | - | 44.30 |
| InstructRAG | 51.2 | 40.7 | - | - | - | - | - | - | 45.95 |
| Auto-RAG | 48.6 | 41.6 | - | - | - | - | - | - | 45.10 |
| *Intermediate Module* | | | | | | | | | |
| Reranker | 58.2 | 42.3 | 36.0 | 30.7 | 63.4 | 41.1 | 52.0 | 48.8 | 46.56 |
| QueryRewrite | 67.2 | 45.9 | 56.8 | 38.5 | 67.4 | 39.2 | 72.0 | 51.8 | 54.85 |
| SKR-KNN | 65.6 | 44.1 | 57.8 | 37.8 | 66.6 | 41.7 | 71.2 | 52.5 | 54.66 |
| SlimPLM | 60.8 | 44.7 | 47.2 | 35.2 | 54.8 | 40.0 | 68.6 | 53.7 | 50.63 |
| C-3PO (Ours) | **72.8** | **50.0** | **61.0** | **41.6** | **71.6** | **47.2** | **74.6** | **55.4** | **59.28** |
| | ↑5.6 | ↑3.7 | ↑3.2 | ↑2.4 | ↑4.2 | ↑3.4 | ↑2.6 | ↑1.7 | ↑4.22 |

## 6.3. Analysis of Plug-and-Play Proxy

In this section, we conduct a comprehensive investigation of performance across three out-of-distribution (OOD) dimensions, including OOD datasets, retrieval systems, and LLM servers. This analysis aims to demonstrate that our proxy is a plug-and-play module with superior generalization capabilities. In Table 2, we assess its modularity and generalization by introducing two recent and challenging OOD datasets: FreshQA (FQA) (Vu et al., 2024) and MultiHop-RAG (M-RAG) (Tang & Yang, 2024). Additionally, we replace the retriever with the Google search engine (Schmidt, 2014) and experiment with different LLM servers.

**First**, LLM fine-tuning approaches exhibit notably inferior performance compared to the standard RAG. This significant degradation suggests that directly fine-tuning LLMs, while potentially effective for specific tasks, may compromise the inherent generalization capabilities and lead to subpar OOD performance. **Second**, while intermediate-module methods maintain competitive performance, their focus on optimizing individual tasks may compromise their robustness. In contrast, our C-3PO holistically optimizes all communication tasks of the entire RAG pipeline through multi-agent collaboration, effectively aligning the retriever and LLM while preserving their inherent generalization capabilities. This enables C-3PO to achieve superior generalization across all OOD settings, consistently outperforming existing approaches with a large margin, 4.22% over the best performing baseline on average. **Finally**, even all three dimensions are OOD, C-3PO exhibits robust performance across different LLM servers (Qwen2-72B, Qwen2-7B, Llama3.3-

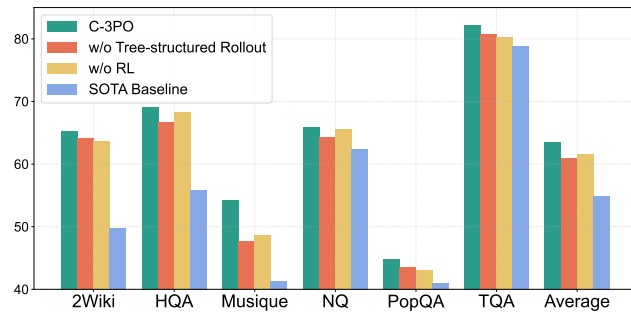

*Figure 2.* Ablation Study.

70B, and GPT-4o-mini), with consistent improvements ranging from 1.7% to 5.6%. This platform-agnostic performance demonstrates the plug-and-play capability of our method, enabling seamless integration with various retrievers and LLM servers without requiring any modifications.

## 6.4. Ablation Study

**Ablation on Training Paradigm.** To thoroughly evaluate the effectiveness of different components in our training process, we conduct comprehensive ablation studies across six in-domain datasets. Specifically, we examine the following variants: (1) "w/o Tree-structured Rollout": A variant without the tree-structured rollout and Monte Carlo credit assignment, meaning that we directly optimize each agent using the system-level reward (a single trajectory). (2) "w/o RL": The performance in the supervised warm-up phase. (3) "SOTA Baseline": The strongest baseline of each dataset.

The experimental results reveal several key findings. **First**, removing the tree-structured rollout (and Monte Carlo credit

Table 3. Ablation Study on Collaborative Strategy.

|                | 2Wiki | PopQA | FQA  | M-RAG |
|----------------|-------|-------|------|-------|
| C-3PO          | 65.2  | 44.8  | 72.8 | 50.0  |
| [No Retrieval] | 41.6  | 24.3  | 58.2 | 42.2  |
| [Retrieval]    | 64.7  | 44.6  | 68.8 | 48.3  |
| [Planning]     | 65.5  | 46.9  | 74.8 | 50.4  |

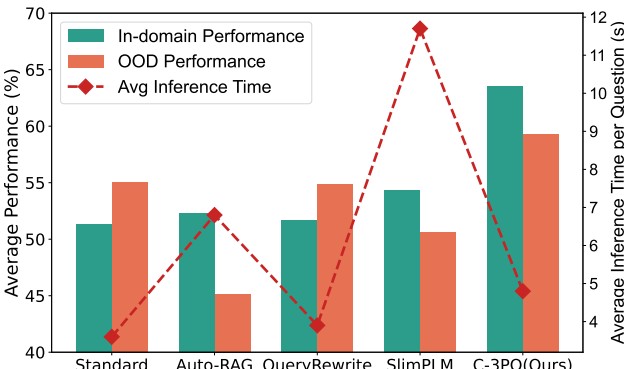

Figure 3. Performance and Efficiency Comparison.

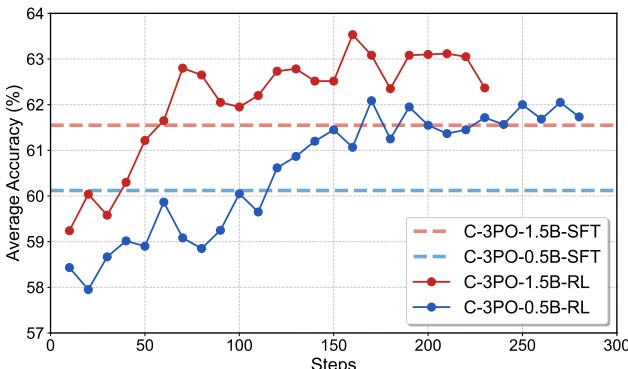

Figure 4. Average Accuracy of C-3PO during RL training.

assignment) leads to unstable performance during the RL phase, occasionally degrading below the supervised warm-up model. This degradation can be attributed to the direct use of system-level rewards as supervised signals for all agents, which fails to accurately assess individual agent contributions and may mask detrimental actions within successful trajectories. In contrast, our Monte Carlo credit assignment mechanism enables reward allocation in probabilistic expectation through tree-structured exploration, ensuring that each agent receives appropriate feedback for its specific actions. **Second**, comparing with the supervised warm-up model ("w/o RL"), our C-3PO achieves substantial improvements across all datasets. This performance boost demonstrates that end-to-end RL optimization effectively aligns the behaviors of multiple agents towards the system-level objectives, going beyond the limitations of supervised learning that only optimizes for local agents. The improvement is particularly significant on challenging datasets like 2Wiki (+1.6%), Musique (+4.5%), PopQA (+1.7%), and TQA (+2.0%), where sophisticated coordination among agents is crucial for task success.

**Ablation on Collaborative Strategy.** To better understand the effectiveness of our collaborative strategy, we conduct ablation studies on OOD datasets by forcing C-3PO to consistently use a fixed strategy for all questions, as shown in Table 3. Notably, [No Retrieval] only utilizes the inherent knowledge of LLMs exhibit the lowest performance, which is suited for addressing straightforward problems. We observe that the [Planning] strategy consistently achieves the best performance among other strategies, which is reasonable given its more sophisticated reasoning process and higher inference cost. Moreover, even our simple [Retrieval] strategy significantly outperforms other baselines shown in Table 2, demonstrating the effectiveness of the retrieval-filter capability in our C-3PO.

### 6.5. Detailed Analysis

**Inference Efficiency Analysis.** To investigate the inference efficiency of C-3PO, we compare both the performance and inference cost across different methods, as illustrated in Figure 3. Our approach achieves superior performance (+9.2% for in-domain and +4.2% for OOD scenarios) while maintaining a reasonable inference time of 4.8s per question. Although slightly slower than the Standard RAG

method (3.6s), C-3PO yields significant performance gains across both in-domain and out-of-generation evaluations. Furthermore, C-3PO outperforms most methods, such as AutoRAG (Yu et al., 2024a) and SlimPLM (Tan et al., 2024), in both efficiency and effectiveness. This demonstrates that C-3PO achieves an optimal balance between performance and computational efficiency.

**Training Dynamics in RL.** Figure 4 depicts the average performance trajectory of our C-3PO across six in-domain benchmarks throughout the RL training process. The results demonstrate consistent and stable improvement in accuracy for both model (C-3PO-1.5B and C-3PO-0.5B) during RL training. The final accuracy of C-3PO-1.5B (63.53%) surpasses that of C-3PO-0.5B (62.08%), suggesting that model capacity plays a role in the ultimate performance ceiling. Both models exhibit rapid improvement during the initial training phase and eventually outperform the SFT model. This significant improvement highlights the effectiveness of our tree-structured multi-agent optimization framework in optimizing multi-agent collaborative systems over time.

### 6.6. Performance on HLE

We evaluate our C-3PO on Humanity's Last Exam (HLE) (Phan et al., 2025), a recently released challenging benchmark. We use the OOD Retrieval (Google Search) and LLM Server (Qwen2.5-72B-Instruct (Yang et al., 2024b))

Table 4. Main Results on Humanity's Last Exam (text-only questions, evaluated with official prompt (Phan et al., 2025)).

| Method | Humanity's Last Exam | | | | | | | | |
|---|---|---|---|---|---|---|---|---|---|
| | Bio/Med | Chem. | CS/AI | Engineering | Humanities | Math | Physics | Other | Avg. |
| *Proprietary Models (For Reference)* | | | | | | | | | |
| OpenAI Deep Research | - | - | - | - | - | - | - | - | 26.60 |
| Deepseek R1 | - | - | - | - | - | - | - | - | 8.54 |
| o1 | - | - | - | - | - | - | - | - | 7.75 |
| GPT-4o | - | - | - | - | - | - | - | - | 2.32 |
| *Open-Source Models* | | | | | | | | | |
| Qwen2.5-7B | 5.42 | 3.00 | 1.76 | 3.22 | 4.66 | 3.58 | 1.98 | 4.00 | 3.52 |
| Qwen2.5-72B | **11.31** | 6.00 | 1.76 | 1.61 | **7.25** | 3.07 | **3.96** | 2.28 | 4.27 |
| C-3PO (Ours) | 9.95 | **7.00** | **4.86** | **9.67** | 4.66 | **5.43** | 3.46 | **5.71** | **5.79** |

Table 5. Comparative Analysis between C-3PO-ICL and C-3PO-RL.

| Method | Proxy | 2Wiki | HQA | Musique | NQ | PopQA | TQA | Average | Efficiency |
|---|---|---|---|---|---|---|---|---|---|
| C-3PO-ICL | Qwen2-72B | 54.1 | 62.5 | 45.5 | 63.4 | 45.7 | 82.9 | 59.01 | 10.7s |
| C-3PO-RL | Qwen2-1.5B | 65.2 | 69.0 | 54.2 | 65.9 | 44.8 | 82.1 | 63.53 | 4.8s |

as our system components. As shown in Table 4, our model achieves an average score of 5.79%, demonstrating a significant improvement over its base model Qwen2.5-72B-Instruct (4.27%). Notably, our model surpasses several proprietary models like GPT-4o (2.32%) and approaches the performance of o1 (7.75%), highlighting the effectiveness of our method in narrowing the gap between open-source and proprietary models.

### 6.7. More Analysis of C-3PO-ICL and C-3PO-RL

We further conduct a comprehensive comparison between C-3PO-ICL (Qwen2-72B-Instruct) and C-3PO-RL (Qwen2-1.5B), where C-3PO-ICL is used to generate seed data through rejection sampling in our supervised warm-up phase. Our experimental results, as presented in Table 5, reveal several important findings. First, C-3PO-ICL demonstrates remarkable performance, surpassing all baseline methods across different datasets (as shown in Table 1 and Table 2). This result validates the effectiveness of our framework, where collaboration among multiple agents enables effective alignment of the LLM and the retriever. However, this approach faces practical limitations due to substantial inference overhead from multiple LLM queries, making it less suitable for efficient responses and edge deployment.

To address these limitations, we introduce a compact proxy that significantly reduces computational requirements while maintaining framework effectiveness. Our analysis reveals that while C-3PO-ICL performs well overall, it may not

achieve optimal performance on more challenging tasks (e.g., 2Wiki, HotpotQA, and Musique). Through reinforcement learning, we further optimize individual agent capabilities, leading to substantial improvements on the complex tasks. In conclusion, our proxy-centric alignment framework demonstrates strong performance across both variants. While C-3PO-ICL showcases the framework's effectiveness through few-shot learning, C-3PO-RL offers practical advantages through reduced computational requirements and enhanced performance on challenging tasks. Our C-3PO-RL successfully aligns the retriever and LLM without modifying either component, while facilitating edge deployment and robust performance across diverse scenarios.

## 7. Conclusion

In this paper, we presented C-3PO, a proxy-centric alignment framework that bridges retrievers and LLMs through a lightweight multi-agent system. By leveraging MARL with our proposed tree-structured rollout and Monte Carlo credit assignment, our framework effectively optimizes multiple specialized agents toward the system-level performance without modifying existing RAG components. Extensive experiments demonstrate C-3PO's superior performance and strong generalization capability across different out-of-distribution datasets, retrievers, and LLMs, establishing it as a practical plug-and-play solution for RAG systems.

## Acknowledgements

This work was done during an internship at Tongyi Lab and was supported by Alibaba Research Intern Program. This work was partially supported by National Natural Science Foundation of China under Grant No. 92470205 and 62222215.

## Impact Statement

The advancements in Retrieval-Augmented Generation (RAG) systems, such as those proposed in this work, hold significant potential for diverse applications. By enhancing modularity, generalization, and plug-and-play capabilities, these systems can empower applications in edge deployment at relatively low computational cost. However, the deployment of RAG systems still poses risks that need to be carefully considered, despite our efforts to mitigate them. These systems heavily rely on external retrieval sources, which may include biased or unreliable data, leading to the propagation of misinformation. In high-stakes applications like medical or legal advice, incorrect or incomplete retrieval could have severe consequences. Although our proxy is not directly responsible for the content retrieved by the retriever, future research can focus on improving alignment in fairness, robustness, and transparency during information filtering.

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

# A. More Implementation Details

## A.1. RL Training Process

Having obtained the credit rewards that reflect each agent's contribution, we develop an optimization framework to guide end-to-end training across all agents. The key idea is to use these credit signals for optimizing the collaborative behavior of the entire system. The optimization objective for our multi-agent system can be formulated as maximizing the expected credit rewards:

$$\mathcal{J}(\theta) = \mathbb{E}_{\tau \sim \pi_\theta} \left[ \sum_{i \in \mathcal{N}} \sum_t r_{\text{credit}}(s_t^i, a_t^i) \right] \tag{7}$$

Since each agent's action is a sequence of tokens, we decompose this optimization using Proximal Policy Optimization (PPO) (Schulman et al., 2017; Yuan et al., 2023a; Zhu et al., 2024) as follows:

$$\mathcal{L}_{\text{C-3PO}} = \sum_{i \in \mathcal{N}} \mathcal{L}_{\text{PPO}}^i(\theta, \phi) \tag{8}$$

Specifically, for each agent $i$, we define:

$$\mathcal{L}_{\text{CLIP}}^i(\theta) = \mathbb{E}_{\tau \sim \pi_\theta} \left[ \sum_t \sum_m \min \left( r_{t,m}^i(\theta) \hat{A}_{t,m}^i, \text{clip}(r_{t,m}^i(\theta), 1 - \epsilon, 1 + \epsilon) \hat{A}_{t,m}^i \right) \right] \tag{9}$$

where $r_{t,m}^i(\theta) = \frac{\pi_\theta(a_{t,m}^i | s_{t,m}^i)}{\pi_{\theta_{\text{old}}}(a_{t,m}^i | s_{t,m}^i)}$ is the probability ratio, $s_{t,m}^i$ represents the concatenation of current state and the first $m - 1$ tokens in the action sequence for agent $i$ at time step $t$, and $a_{t,m}^i$ denotes its $m$-th token. We compute the advantage estimate using GAE (Schulman et al., 2016): $\hat{A}_{t,m}^i = \sum_{l=0}^{M-m-1} (\gamma \lambda)^l \delta_{t,m+l}^i$, where $M$ is the token length of the action sequence.

To estimate state values across the multi-agent system, we employ a centralized state-value function $V_\phi$ that takes each agent's state $s_{t,m}^i$ as input. The value function is optimized to minimize the mean squared error (Lowe et al., 2017):

$$\mathcal{L}_V^i(\phi) = \mathbb{E}_{\tau \sim \pi_\theta} \left[ \sum_t \sum_m (V_\phi(s_{t,m}^i) - \hat{G}_{t,m}^i)^2 \right] \tag{10}$$

where $\hat{G}_{t,m}^i = \hat{A}_{t,m}^i + V_\phi(s_{t,m}^i)$ is the empirical return. The final optimization objective combines the policy and value losses:

$$\mathcal{L}_{\text{PPO}}^i(\theta, \phi) = \mathcal{L}_{\text{CLIP}}^i(\theta) + c_v \mathcal{L}_V^i(\phi) \tag{11}$$

where $c_v$ controls the weight of the value loss. This joint objective enables end-to-end training of both policy and value networks across all agents.

## A.2. Implementation Details

**Supervised Warm-up Phase:** We utilize `Llama-Factory` (Zheng et al., 2024) as our training framework for the initial supervised fine-tuning phase. The detailed hyper-parameters for this phase are presented in Table 6.

**Reinforcement Learning Phase:** For the RL training phase, we adopt `OpenRLHF` (Hu et al., 2024) as our primary training framework, coupled with `VLLM` (Kwon et al., 2023) inference engine. The complete set of RL training hyper-parameters is detailed in Table 7. To initialize both the policy and value models, we leverage the model obtained after one epoch of supervised fine-tuning, with the language model head replaced by a value head for the value model.

**Inference Phase:** For the deployment of our system, we establish a comprehensive infrastructure that integrates multiple components:

- **Retriever Server:** We construct our retrieval server using the 2018 Wikipedia dump (Yang et al., 2018) as the primary knowledge source. We employ contriever-msmarco (Izacard et al., 2022) as our dense retriever for efficient and effective document retrieval. Our inference code also supports Google search engine (Schmidt, 2014) as the retriever server.

Table 6. Key hyperparameters in the supervised warm-up phase.

| Hyperparameter | Value |
|---|---|
| Learning Rate | 4e-5 |
| Batch size | 512 |
| #Epochs | 3 |
| Optimizer type | AdamW (Loshchilov & Hutter, 2019) |
| Chat template | Qwen (Yang et al., 2024a) |
| Base model | Qwen2-1.5B or Qwen2-0.5B (Yang et al., 2024a) |
| Cutoff length | 4096 |
| Warmup ratio | 0.03 |
| LR scheduler type | Cosine |

Table 7. Key hyperparameters in the RL phase.

| Hyperparameter | Value |
|---|---|
| Learning Rate of Policy model | 5e-7 |
| Learning Rate of Value model | 5e-6 |
| Batch size | 1024 |
| KL Coefficient | 0.005 |
| Optimizer type | Adam |
| Prompt max len | 4096 |
| Generate max len | 2048 |
| Maximal depth | 13 |
| LR scheduler type | Cosine |

- **LLM Service:** We integrate `SGLang`[4] as our LLM server, which provides compatibility with various state-of-the-art language models, including Qwen2-72B-Instruct (Yang et al., 2024a) and Llama3.3-70B-Instruct (Dubey et al., 2024). Moreover, we also support GPT series models[5].

- **Inference Optimization:** Our implementation supports two high-performance inference engines: `SGLang` and `VLLM`, allowing users to optimize for different deployment scenarios and hardware configurations.

This modular architecture ensures both flexibility in model selection and efficiency in deployment, while maintaining robust performance across different configurations.

### A.3. Dataset Details

Table 8. Training Dataset Statistics.

| Data Name | Multi-Hop | | | Single-Hop | | | Total |
|---|---|---|---|---|---|---|---|
| | 2WikiMultiHopQA | HotpotQA | Musique | Natural Questions | PopQA | TriviaQA | |
| **Raw Data Size** | 167,454 | 90,447 | 19,938 | 79,169 | 12,868 | 78,785 | 448,661 |
| **Our Train Data Size** | 6,000 | 6,000 | 6,000 | 6,000 | 6,000 | 6,000 | 36,000 |
| **Sampling ratio** | 3.5% | 6.6% | 30.1% | 7.5% | 46.6% | 7.6% | 8.02% |

**In-domain Datasets.** As shown in Table 8, we conduct extensive in-domain experiments on three single-hop and three multi-hop datasets. For each dataset, we randomly sampled 6,000 instances as the training set, with sampling ratios detailed in Table 8. Overall, we utilize only 8% of the original data as the training set. For the in-domain test sets, we randomly sampled 1,000 instances as the test set.

---

[4] https://docs.sglang.ai/
[5] We use `gpt-4o-mini-2024-07-18` in our out-of-generalization experiments.

*Table 9.* Out-of-generalization Dataset Statistics.

|  | **FreshQA** | **Multihop-RAG** |
|---|---|---|
| Data Size | 500 | 2556 |

**Out-of-generalization Datasets.** To comprehensively evaluate the plug-and-play capability of our C-3PO in out-of-distribution generalization scenarios, we introduce two recent challenging datasets: FreshQA (Vu et al., 2024) and Multihop-RAG (Tang & Yang, 2024). The statistics of the OOD datasets are summarized in Table 9.

### A.4. Overall Algorithm

In this section, we present the inference process of C-3PO for reference, as shown in the Algorithm 1.

---

**Algorithm 1** Inference Process of our C-3PO

---

**input** question $q$, the retrieval server (**Retriever**), the LLM server (**LLM**), the proxy model in our C-3PO $\pi$, instruction for different agent (Reasoning Router, Information Filter, and Decision Maker).

**output** The Answer.

1: $a^1 \leftarrow \pi(q, \text{instruction}_1)$ {Reasoning Router agent}
2: **if** $a^1$==[No Retrieval] **then**
3:     Answer $\leftarrow$ **LLM**$(q)$ {Direct Answering Strategy, if $q$ does not require retrieval}
4: **else if** $a^1$==[Retrieval]`<query content>` **then**
5:     docs $\leftarrow$ **Retrieval**(`<query content>`)
6:     selected docs$(a^2) \leftarrow \pi(q, \text{docs}, \text{instruction}_2)$ {Information Filter agent}
7:     Answer $\leftarrow$ **LLM**$(q, \text{selected docs})$ {Single-pass Strategy, if $q$ requires retrieval and is simple question}
8: **else**
9:     Accumulated_docs $\leftarrow \emptyset$
10:     Roadmap $\leftarrow$ **LLM**$(q)$ {$a^1 ==$ [Planning]}
11:     $a^3 \leftarrow \pi(q, \text{Roadmap}, \text{Accumulated\_docs}, \text{instruction}_3)$ {Decision Maker agent}
12:     **while** $a^3 \neq$ [LLM] **do**
13:         docs $\leftarrow$ **Retrieval**(`<subquery content>` in $a^3$)
14:         selected docs$(a^2) \leftarrow \pi(q, \text{docs}, \text{instruction}_2)$ {Information Filter agent}
15:         Accumulated_docs $\leftarrow \{\text{Accumulated\_docs}\} \cup \{\text{selected docs}\}$
16:         $a^3 \leftarrow \pi(q, \text{Roadmap}, \text{Accumulated\_docs}, \text{instruction}_3)$ {Decision Maker agent}
17:     **end while**
18:     Answer $\leftarrow$ **LLM**$(q, \text{Accumulated\_docs})$ {Multi-step Reasoning Strategy, if $q$ requires retrieval and is complex}
19: **end if**

---

## B. Instructions and State Transition Function

### B.1. Instructions for Each Agent

In this section, we details the state space and action space fo each agent in our C-3PO.

**Reasoning Router.** The Reasoning Router agent operates with state space $\mathcal{S}^1 = \{q\}$, where $q$ represents the input question. This agent is responsible for determining whether retrieval is necessary for the given question and assessing the question complexity when retrieval is needed. For a question that does not require retrieval, this agent outputs [No Retrieval]. If the retrieval is needed, the agent outputs one of the following actions based on the complexity of the question $q$: for simple questions requiring retrieval: [Retrieval]`<query content>`, initiating a single-pass retrieval-filter loop, where `<query content>` defines the space of possible queries; for complex questions: [Planning], triggering the multi-step reasoning strategy. The specific examples are as follows:

---

**Reasoning Router**

**Instruction for Reasoning Router**

You are an intelligent assistant tasked with evaluating whether a given question requires further information through retrieval or needs planning to arrive at an accurate answer. You will have access to a large language model (LLM) for planning or answering the question and a retrieval system to provide relevant information about the query.

Instructions:
1. **Evaluate the Question**: Assess whether a precise answer can be provided based on the existing knowledge of LLM. Consider the specificity, complexity, and clarity of the question.
2. **Decision Categories:**
- If the question is complex and requires a planning phase before retrieval, your response should be:
`[Planning]`
- If the question requests specific information that you believe the LLM does not possess or pertains to recent events or niche topics outside LLM's knowledge scope, format your response as follows:
`[Retrieval]` `'YOUR QUERY HERE'`
- If you think the LLM can answer the question without additional information, respond with:
`[No Retrieval]`
3. **Focus on Assessment**: Avoid providing direct answers to the questions. Concentrate solely on determining the necessity for retrieval or planning.

**State of Reasoning Router**
Now, process the following question:

Question: {question}

**Output (All possible Actions) of Reasoning Router**
% For No Retrieval
`[No Retrieval]`
% For Retrieval
`[Retrieval]<query content>` (for simple questions)
`[Planning]` (for complex questions)

---

**Information Filter.** The state space of Information Filter consists of the question $q$, the retrieved documents, and the current objective (if in `[Planning]` mode), i.e., $\mathcal{S}^2 = \{q, \text{retrieved documents}\}$ for single-pass strategy (`Retrieval<query content`), or $\mathcal{S}^2 = \{q, \text{retrieved documents}, \text{current objective}\}$ for multi-step reasoning strategy (`[Planning]`).

---

**Information Filter**

**Instruction for Information Filter**

You are an intelligent assistant tasked with analyzing the retrieved documents based on a given question and the current step's objectives. Your role is to determine the relevance of each document in relation to the question and the specified objectives.

Instructions:
1. **Analyze Relevance**: Evaluate each document whether it aligns with the objectives of the current retrieval step and contains a direct answer to the question.
2. **Thought Process**: Provide a brief analysis for each document, considering both the answer content and the retrieval objectives.
3. **Filter Documents**: After your thought process, generate a list of document indices indicating which documents to retain.

**State of Information Filter**
Now, process the following question:

Current step's objectives: {objective} (only for `[Planning]` mode)

Question: {question}

Documents: {documents}

**Output of Information Filter**
Thought: <Analysis of each documents>

---

> Action: [<Selected document IDs>]

**Decision Maker.** The Decision Maker agent operates with state space $\mathcal{S}^3 = \{q, \text{Accumulated Documents}, \text{Roadmap}\}$. Based on the current state, this agent outputs one of two possible actions: `[Retrieval]<subquery content>` (requesting additional retrieval-filtering loop through the sub-query) or `[LLM]` (passing all accumulated documents to LLM for generating the final answer).

---

**Decision Maker**

**Instruction for Decision Maker**
You are an intelligent assistant tasked with determining the next appropriate action based on the provided existing documents, plan, and question. You have access to a large language model (LLM) for answering question and a retrieval system for gathering additional documents. Your objective is to decide whether to write a query for retrieving relevant documents or to generate a comprehensive answer using the LLM based on the existing documents and plan.

Instructions:
1. **Evaluate Existing Documents**: Assess the existing documents to determine if it is sufficient to answer the question.
2. **Follow the Plan**: Understand the next steps outlined in the plan.
3. **Decision Categories:**
- If the existing documents is insufficient and requires additional retrieval, respond with:
`[Retrieval]` `YOUR QUERY HERE`
- If the existing documents is adequate to answer the question, respond with:
`[LLM]`
4. **Focus on Action**: Do not answer the question directly; concentrate on identifying the next appropriate action based on the existing documents, plan, and question.

**State of Decision Maker**
Now, process the following question:

Existing Documents: {accumulated documents}

Roadmap: {roadmap}

Question: {question}

**Output of Decision Maker**
Thought: [Your analysis for current situation (need retrieval for additional informations or use LLM to answer)]
Action: [Your decision based on the analysis (`[Retrieval]<subquery content>` or `[LLM]`)]

---

## B.2. State Transition Function

Given a state $s_t^i$ and an action $a_t^i$ in each agent $i \in \mathcal{N}$, the transition function $\mathcal{T}$ in our framework is deterministic. Based on the three collaborative strategies introduced in Section 4.2, the state transitions are defined as follows:

**Direct Answering Strategy** (`[No Retrieval]`): In this strategy, the LLM directly generates the answer without retrieval, resulting in no state transitions between agents.

**Single-pass Strategy** (`[Retrieval]<query content>`): This strategy involves a state transition between the Reasoning Router and Information Filter agents:

$$\mathcal{T} : \mathcal{S}^1 = \{q\} \times \mathcal{A} = \{\texttt{[Retrieval]<query content>}\} \xrightarrow{\text{retrieval}} \mathcal{S}^2 = \{q, \text{retrieved documents}\} \qquad (12)$$

where $\mathcal{S}^1$ represents the initial state with the question $q$, and $\mathcal{S}^2$ represents the state for the Information Filter agent after retrieval. The Information Filter is responsible for filtering the helpful documents based on $\mathcal{S}^2$.

**Multi-Step Reasoning Strategy** (`[Planning]`): This strategy involves multiple state transitions in a cyclic manner:

- Reasoning Router → Decision Maker:

$$\mathcal{T} : \mathcal{S}^1 = \{q\} \times \mathcal{A} = \{\texttt{[Planning]}\} \xrightarrow{\texttt{[planning]}} \mathcal{S}^3 = \{q, \text{Accumulated Documents}, \text{Roadmap}\}, \qquad (13)$$

where the roadmap is generated by the LLM and the accumulated documents is empty for initial step.

- Decision Maker → Information Filter:

$$\mathcal{T} : \mathcal{S}^3 = \{q, \text{Accumulated Documents}, \text{Roadmap}\} \times \mathcal{A} = \{\texttt{[Retrieval]<subquery content>},$$

$$\text{current objective}\} \xrightarrow{\text{retrieval}} \mathcal{S}^2 = \{q, \text{retrieved documents}, \text{current objective}\}, \qquad (14)$$

where the current objective is generated by the Decision Maker agent in $\mathcal{S}^3$.

- Information Filter → Decision Maker:

$$\mathcal{T} : \mathcal{S}^2 = \{q, \text{retrieved documents}, \text{current objective}\} \times \mathcal{A} = \{\text{Selected Documents}\}$$

$$\xrightarrow{\text{filter}} \mathcal{S}^3_{\text{new}} = \{q, \text{Updated Accumulated Documents}, \text{Roadmap}\}. \qquad (15)$$

This retrieval-filter loop between the Decision Maker agent and the Information Filter agent continues until the Decision Maker outputting [LLM] or a termination condition is met. The state transitions in our C-3PO are deterministic and well-defined, ensuring consistent behavior across the multi-agent system.

## C. Additional Experimental Results

### C.1. More Analysis in RL

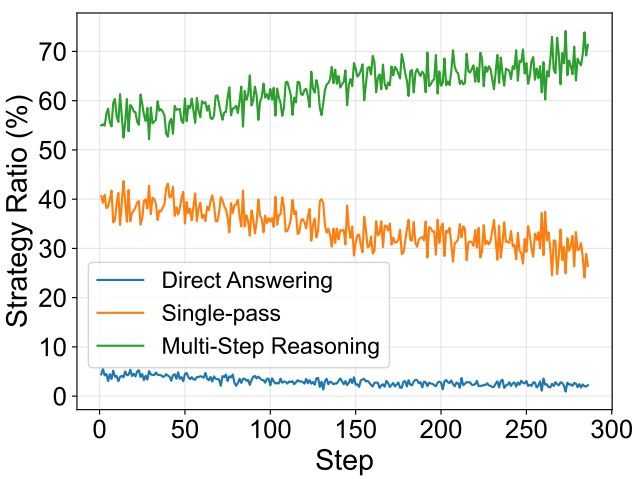

*Figure 5.* Strategy Ratio in RL training process.

**Strategy Ratio during RL Training Process.** As introduced in the Section 4.2, our C-3PO incorporates three distinct strategies: Direct Answering Strategy ([No Retrieval]), Single-pass Strategy ([Retrieval]<query content>), and Multi-Step Reasoning Strategy ([Planning]), each designed for different question complexities. Figure 5 reveals how C-3PO dynamically adapts its strategy selection during the RL training process.

The evolution of strategy ratios shows a clear trend: the Multi-Step Reasoning Strategy gradually dominates the decision space, stabilizing at approximately 60-70%, while the Single-pass Strategy decreases to around 30%. The Direct Answering Strategy maintains a consistent but low ratio of about 5%. This distribution pattern offers several insights into our framework's learning behavior: **First**, the limited use of Direct Answering Strategy aligns with our experimental findings in Table 1, confirming that solely relying on the model's inherent knowledge is insufficient for complex question-answering tasks. **Second**, the substantial proportion of Single-pass Strategy usage demonstrates our C-3PO's ability to identify scenarios where simple external information retrieval suffices. **Most notably**, the increasing preference for Multi-Step Reasoning Strategy indicates that our C-3PO recognizes the importance of multi-step reasoning in handling complex queries effectively. These learned ratios demonstrate that our framework effectively develops a balanced strategy selection mechanism. By dynamically choosing appropriate strategies based on question complexity, our C-3PO achieves a balance between computational efficiency and reasoning capability, making it well-suited for real-world applications.

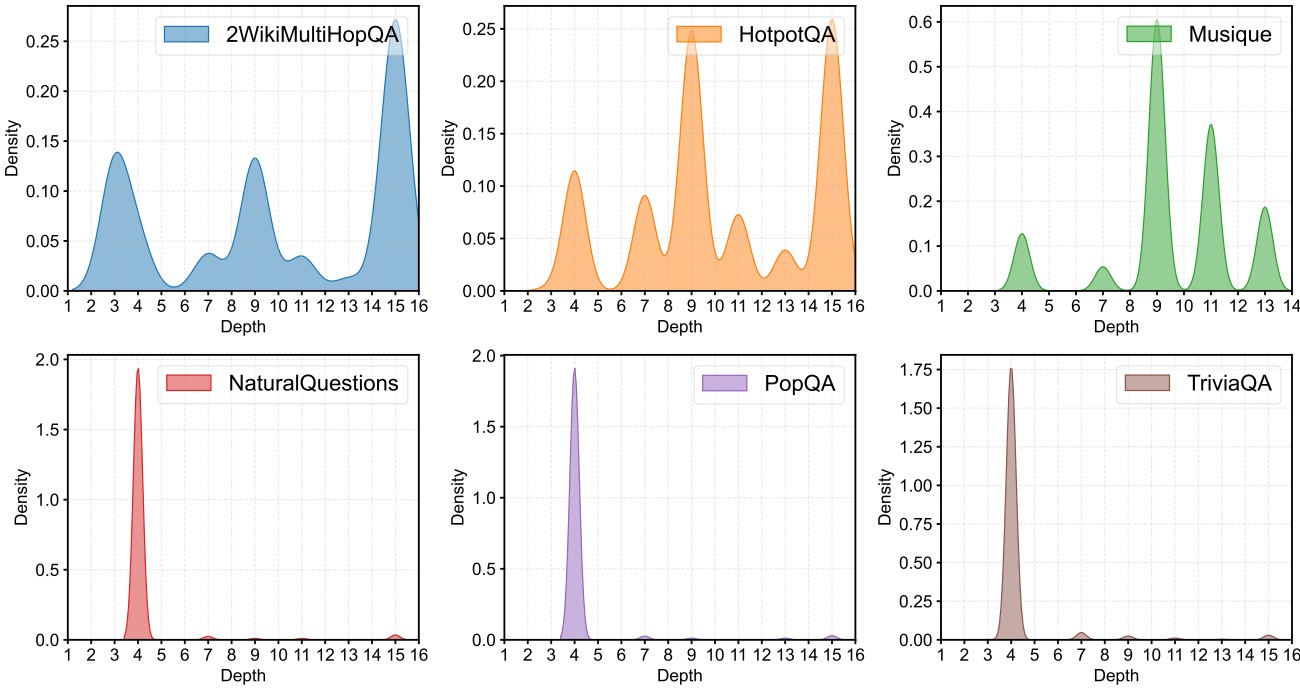

*Figure 6.* Depth Distribution in Test set.

**Depth Distribution.** Figure 6 presents the depth distribution of reasoning processes across different datasets, revealing distinct patterns that align with the inherent complexity of each task. We observe three clear categories of reasoning depth requirements: **(1) Simple Complexity (Depth 3-5):** Datasets like NaturalQuestions, PopQA, and TriviaQA show concentrated distributions around depths 3-5, indicating that most questions in these datasets can be effectively addressed with the Direct Answering Strategy (`[No Retrieval]`) and Single-pass Strategy (`[Retrieval]<query content>`). This aligns with the nature of these datasets, which primarily contain straightforward factual questions. **(2) Mixed Complexity:** HotpotQA and 2WikiMultiHopQA exhibit multiple peaks in the depth distribution, with notable concentrations around depths 3-4 and depths 9-15, indicating a diverse range of question complexity. This bimodal distribution suggests that while some questions require simple reasoning steps, others need more complex reasoning chains. **(3) Complex Complexity:** Musique displaies broader distributions with significant density at higher depths (9-13), particularly pronounced in their rightward skew. Musique's distribution is notably spread across higher depths, consistent with its design for multi-step reasoning questions.

These distributions validate our framework's adaptive capability in handling queries of varying complexity. The framework naturally adjusts its reasoning depth based on task requirements, demonstrating efficient resource utilization while maintaining the ability to perform deep reasoning when necessary.

## D. Additional Prompts

In this section, we supplement additional prompts based on Appendix B.

### D.1. Roadmap

In the multi-step reasoning strategy, we introduce an LLM-generated roadmap as high-level guidance for our proxy. The specific prompt and example are as follows:

---

**An example of Roadmap**

Prompt for Roadmap
You are an expert assistant tasked with analyzing the following question and formulating a detailed plan. You will utilize a retrieval system to gather relevant information in your planning. Your goal is to analysis the question and provide a structured sequence of actions to address it effectively.

Instructions:
1. **Question Analysis**: Identifying the core components of the question. Determine what key information we currently know and what additional information is needed through retrieval.
2. **Step By Step Planning**: Develop a detailed plan step by step. Focus on the planning process rather than providing direct answers.
3. **Focus on Planning**: Keep your response clear and structured, concentrating solely on the analysis and planning aspects.

Now, process the following question:

Question: {question}

Example of generated roadmap
(Take `What nationality is the director of film The Caper Of The Golden Bulls?` as an example)
To answer the question, we need to find information about the director of the film "The Caper of the Golden Bulls." Then we should determine which nationality is the director born using the retrieval.
Step 1: Retrieve the relevant documents that mention the film 'The Caper of the Golden Bulls.'
Step 2: Identify the director of the film from the retrieved documents.
Step 3: Retrieve the relevant information about 'Which nationality is the director born¿.
Step 4: Provide the answer based on the retrieved information.

## D.2. Evaluation

In our experiments, we found that traditional evaluation metrics such as Exact Match (EM) are often inaccurate, as they strictly require identical generated answers. To address this issue, following previous work (Zheng et al., 2023; Vu et al., 2024), we leverage an LLM to assess answer correctness by comparing the predicted answer with the ground truth. The specific example is as follows:

---

**Prompt of Evaluation**

You are a precise answer validator. Your task is to compare the predicted answer with a set of acceptable correct answers and determine if the prediction matches any of them.

Input format:
Question: [The question text]
Correct Answers: [Array or list of acceptable correct answers]
Predicted Answer: [The answer to be evaluated]

Rules:
1. Consider semantic equivalence, not just exact string matching
2. Ignore minor differences in formatting, spacing, or capitalization
3. For numerical answers, consider acceptable margin of error if applicable
4. For text answers, focus on the core meaning rather than exact wording
5. The predicted answer is considered correct if it matches ANY ONE of the provided correct answers
6. The matching can be exact or semantically equivalent to any of the correct answers
7. Return only "True" if the predicted answer is correct, or "False" if it is incorrect.

Now, process the following question:
Question: {question}
Correct Answer: {true_answer}
Predicted Answer: {long_answer}

