# OpenReview forum: "C-3PO: Compact Plug-and-Play Proxy Optimization to Achieve Human-like Retrieval-Augmented Generation"
_ICML.cc/2025/Conference — ICML 2025 poster_

### Official Review · Reviewer_stM2 · 2025-03-12

**Overall Recommendation:** 4

**Summary:**

This paper presents a novel proxy-centric framework for addressing the alignment challenge in RAG systems. The key innovation lies in its introduction of a lightweight multi-agent system that mediates between retrievers and LLMs without requiring modifications to either component. The framework is inspired by human search behavior and implements three specialized agents that work collaboratively to optimize the RAG pipeline. The key technical contributions of C-3PO include a proxy-centric alignment architecture that maintains plug-and-play flexibility, an efficient multi-agent system design, and a tree-structured rollout approach for multi-agent reinforcement learning that enables effective reward credit assignment. Through extensive experimentation in both in-domain and out-of-distribution scenarios, the authors demonstrate that their approach significantly enhances RAG performance while maintaining generalization capabilities across different retrievers and LLMs.

**Claims And Evidence:**

The authors make several key claims that are well-supported by the presented evidence:
* Claim 1: The proposed proxy-centric framework (C-3PO) effectively bridges retrievers and LLMs while maintaining plug-and-play flexibility.

The authors detailed technical descriptions that clearly distinguish their approach from existing methods to support this claim in Section 1 and 2. The authors demonstrate the plug-and-play flexibility through extensive experiments in both in-domain and out-of-distribution scenarios (Table 1 and Table 2). Evidence appears convincing as they test with unseen retrievers and LLMs to validate generalization capabilities.

* Claim 2: The tree-structured rollout mechanism and Monte Carlo credit assignment effectively optimize multi-agent coordination.

The authors provide a theoretical foundation in Section 5 with detailed mathematical formulation. The effectiveness of this approach is empirically validated through comprehensive ablation studies in Section 6.4, which quantitatively demonstrate its advantages over alternatives. The design are well-motivated and the consistent performance improvements across different experimental settings further strengthen this claim.

* Claim 3: The human-inspired multi-agent collaborative system enhances RAG performance.

The evidence for this claim is particularly strong in Section 6.6 and Appendix C, where the authors demonstrate the effectiveness of their approach through in-context learning experiments. Notably, C-3PO-ICL shows impressive performance even without any training, outperforming many baselines from Tables 1 and 2. The detailed case studies and comprehensive analysis across different tasks and scenarios provide convincing support for the benefits of the multi-agent collaborative approach.

**Essential References Not Discussed:**

After a thorough review of the paper's citations and related work section, I did not identify any essential references that are missing from the discussion. The citation coverage appears complete and up-to-date, providing adequate context for understanding the paper's contributions and positioning in the broader research landscape.

**Experimental Designs Or Analyses:**

The experimental design and analyses in this paper are thorough and well-executed. The authors conduct comprehensive experiments across a diverse range of datasets, including both single-hop and multi-hop benchmarks , which effectively validates the model's capability in handling varying complexity levels of tasks.

Particularly noteworthy is their extensive evaluation of out-of-distribution (OOD) generalization across three dimensions: OOD datasets (FreshQA and MultiHop-RAG), different retrieval systems (from Contriever to Google Search), and various LLM servers (from Qwen to GPT-4). This comprehensive OOD evaluation protocol strongly supports their claims about the framework's plug-and-play capability and generalization ability.

The ablation studies are systematic and well-designed. The authors thoroughly examine both the training paradigm and collaborative strategies, providing clear insights into each component's contribution. The comparison of different fixed strategies particularly helps understand the model's behavior. Furthermore, the efficiency analysis comparing both performance and inference cost across different methods demonstrates practical considerations for real-world deployment.

**Methods And Evaluation Criteria:**

* Methods:
The proposed proxy-centric framework makes sense as it addresses the key challenge of aligning retrievers and LLMs without modification. The multi-agent design mimicking human search behavior is intuitive and well-motivated. The use of MARL with the proposed tree-structured rollout is appropriate for optimizing multiple agents towards the system-level objectives. The lightweight design ensures practical applicability while maintaining effectiveness.

* Evaluation criteria:
The evaluation is comprehensive and well-structured. The authors conduct extensive experiments across a diverse range of datasets, including three single-hop datasets (NQ, PopQA, TriviaQA) and three multi-hop datasets (HotpotQA, 2WikiMultihopQA, MuSiQue). The inclusion of FreshQA and MultiHop-RAG as out-of-distribution test sets further demonstrates the model's robustness and adaptability. Furthermore, the authors evaluate C-3PO's plug-and-play and generalization capabilities by testing with previously unseen retrievers and LLMs. This comprehensive evaluation protocol provides strong evidence for the framework's versatility and practical applicability in real-world settings.

**Other Comments Or Suggestions:**

The paper is well-written and clearly structured. The authors have done a thorough job in presenting their ideas and experimental results. The figures and tables are informative and well-organized. I would encourage the authors to explore the framework's capabilities on more challenging tasks (such as HLE) and investigating its potential for broader applications.

**Other Strengths And Weaknesses:**

Strength
1. The proxy-centric alignment framework is innovative, offering a practical solution that enhances RAG systems without modifying existing components. This approach significantly reduces deployment barriers while maintaining strong performance.
2. The multi-agent collaborative system design is elegant and well-motivated, effectively mimicking human search behavior through specialized agents. The lightweight implementation (0.5B/1.5B parameters) demonstrates impressive efficiency.
3. The training methodology combining MARL with tree-structured rollout and Monte Carlo credit assignment is technically sound and novel, effectively addressing the complex challenge of multi-agent optimization.
4. The empirical validation is remarkably comprehensive, demonstrating strong performance across both in-domain scenarios and out-of-distribution settings (datasets, retrievers, and LLMs), convincingly validating the framework's effectiveness and generalization capability.

Weakness
1. While the current evaluation is comprehensive across in-domain and out-of-distribution settings, testing on more challenging benchmarks like Humanity's Last Exam (HLE) would further validate the model's capabilities on highly complex reasoning tasks.
2. The training paradigm currently relies on seed data collection. While this is a practical approach, exploring the possibility of from-scratch RL training (similar to recent advances in RL (Deepseek-R1)) could provide interesting insights into more general training strategies, though this is beyond the scope of the current work.

**Questions For Authors:**

The key points have been covered in the previous sections. I have no additional questions that would substantially impact my evaluation of this work.

**Relation To Broader Scientific Literature:**

This work makes meaningful connections to several important research directions in the broader scientific literature:

First, the work builds upon and extends retrieval-augmented generation (RAG) research. While previous works mainly focus on modifying either retrievers (e.g., REPLUG) or LLMs (e.g., Self-RAG, Auto-RAG), this paper proposes a novel perspective of using a lightweight proxy for alignment, which provides a more practical and efficient solution.

Second, the tree-structured rollout mechanism for multi-agent reinforcement learning builds upon classic MARL literature. This work presents a solution by introducing Monte Carlo credit assignment with tree-structured exploration, advancing the field of multi-agent coordination.

**Theoretical Claims:**

This paper does not make formal theoretical claims requiring rigorous proofs.

---

> ### Author Rebuttal · Authors · 2025-03-31
>
> Dear Reviewer stM2,
>
> Thank you for your thoughtful review and constructive suggestions. We particularly appreciate your recommendations about extending our evaluation to more challenging benchmarks and exploring alternative training strategies. These insights will help strengthen our work. We would like to address your suggestions in detail:
>
> > W1
>
> Thank you for this valuable suggestion about testing on more challenging benchmarks. We agree that evaluation on complex reasoning tasks is crucial for validating our framework's capabilities.
>
> We have conducted additional experiments on Humanity's Last Exam (HLE) text-only questions using Google as the retriever (an out-of-domain search engine for C-3PO):
>
> | LLM                  | Method         | n docs | HLE (text) |
> |----------------------|----------------|:------:|:----------:|
> | Deepseek-R1          | -              | -      |     8.6    |
> | o3-mini (high)       | -              | -      |     14     |
> | Qwen2.5-72B-Instruct | Vanilla LLM    | -      |    4.85    |
> | Qwen2.5-72B-Instruct | Vanilla RAG    |   10   |    5.35    |
> | Qwen2.5-72B-Instruct | C-3PO          |   10   |    6.46    |
> | Qwen2.5-72B-Instruct | C-3PO-Planning |   10   |    6.84    |
>
> The results show that:
> - C-3PO improves performance by 1.11 % compared to vanilla RAG
> - C-3PO-Planning further boosts performance by 1.49%
> - These improvements demonstrate our framework's effectiveness even on highly challenging reasoning tasks with out-of-domain retrieval
>
> We will include these results in our revised manuscript to provide a more comprehensive evaluation of our framework's capabilities.
>
> > W2
>
> Thank you for this insightful suggestion about exploring from-scratch RL training. We agree that this direction, similar to Deepseek-R1's approach, is very interesting and could potentially lead to more general training strategies for multi-agent systems.
>
> While our current warm-up approach helps ensure stable and smooth training in the multi-agent setting, we believe exploring from-scratch training could:
> - Reduce dependency on seed data collection
> - Potentially discover novel agent interaction patterns
> - Lead to more generalizable training strategies
>
> We will include this as an important direction for future research. The challenge of balancing exploration and stability in from-scratch multi-agent RL training presents an exciting opportunity for advancing the field.
>
> We sincerely appreciate your valuable suggestions that have helped us identify important directions for both immediate improvements and future research. Your feedback about evaluation on challenging benchmarks has already led to meaningful additional results. We will incorporate these improvements in our revised manuscript.

---

> > ### Comment · Reviewer_stM2 · 2025-04-03
> >
> > Thanks for your detailed responses. All concerns have been addressed. I decided to maintain my score to accept.

---

> > > ### Author Response · Authors · 2025-04-04
> > >
> > > Thank you very much for your positive feedback and support. We greatly appreciate your time and consideration.

---

### Official Review · Reviewer_Tndf · 2025-03-12

**Overall Recommendation:** 4

**Summary:**

This paper proposes a proxy-centric framework that enhances communication between retrievers and Large Language Models (LLMs) through a lightweight multi-agent system named C-3PO. Unlike the vanilla RAG framework, the proposed framework incorporates multiple specialized LLM agents to manage different stages of the pipeline:
1. Reasoning Router Agent: Evaluates the complexity of the query to determine whether retrieval and reasoning are required. For simple queries, the process proceeds directly to the Information Filter Agent. For complex queries, the system enters a planning mode, engaging all agents collaboratively.
2. Information Filter Agent: Processes and extracts relevant information from the retrieved data.
3. Decision Maker Agent: Identifies the optimal action during the planning mode.

To train the framework, the authors propose a tree-structured rollout mechanism for credit assignment, addressing the issue of sparse rewards, and utilize a PPO training objective. Experiments conducted on multiple QA datasets across various RAG systems, including those with retriever tuning or LLM tuning, demonstrate significant improvements in performance.

**Claims And Evidence:**

Yes

**Essential References Not Discussed:**

No

**Experimental Designs Or Analyses:**

For the RAG baseline involving LLM fine-tuning, the use of Qwen2 to control variables raises concerns about reproducibility. To ensure fairness and simplicity, I think a more straightforward baseline could simply employ retrirever with an instruction-tuned Qwen2-7B server. Instruction tuning is a standard and widely accessible approach compared to the custom fine-tuning proposed in this work, making it a more practical and reproducible baseline for evaluation.

**Methods And Evaluation Criteria:**

Here arises a minor concern regarding the generalization capability beyond QA tasks. The current agent functions and pipeline appear to be QA-oriented, and the evaluation datasets are exclusively focused on QA tasks. It would be beneficial to either explicitly position this work as specific to QA or extend the evaluation to include a broader range of tasks to demonstrate the framework's versatility and applicability beyond question answering.

**Other Comments Or Suggestions:**

No.

**Other Strengths And Weaknesses:**

### Strengths
1. The use of multi-agent systems to handle complex tasks is a highly sought-after approach, and this paper presents a well-designed framework with significant performance improvements.
2. The paper is well-written and easy to follow, making it accessible to a broad audience.

### Weaknesses
1. The designed agent functionality and pipeline appear to be overly specific to QA tasks, limiting the framework's generalizability to other applications.
2. The reported improvements come at a significant cost, including the computational and resource overhead of training these customized agents and the increased complexity during inference.

Despite the inclusion of an Inference Efficiency Analysis to highlight performance trade-offs, the comparison baseline is somewhat outdated and relies on costly methods (e.g., query rewriting). Recent works have focused on more efficient single-dimension improvements for RAG (e.g., reranking [1], drafting [2]), which were omitted in the main experiments and the Efficiency Analysis.

[1] RankRAG: Unifying context ranking with retrieval-augmented generation in llms. NIPS 2024.

[2] Speculative RAG: Enhancing Retrieval Augmented Generation through Drafting. ICLR 2025.

**Questions For Authors:**

1. To better assess the efficiency and the cost of the framework, could you elaborate on the average number of 8B LLM forward passes (from the additional agents) required for each task?

2. In Table 3, the [Planning] module shows limited improvement on 2Wiki, PopQA, and M-RAG, while demonstrating significant improvement on FQA compared to the [Retrieval] module. Could you provide insights into this discrepancy?

3. Do you think the designed multi-agent framework could be applied to broader tasks beyond QA? For example, tasks in [3]. If not, what adjustments would be necessary to made?

[3] Buffer of Thoughts: Thought-Augmented Reasoning with Large Language Models. NIPS 2024.

**Relation To Broader Scientific Literature:**

This work proposes a multi-agent cooperative framework and training method, which extends beyond QA tasks. Its modular design and tree-structured rollout approach offer potential for broader applications with customizable agents and pipelines.

**Theoretical Claims:**

Yes

---

> ### Author Rebuttal · Authors · 2025-03-31
>
> Dear Reviewer Tndf,
>
> Thank you for your thorough and constructive review. We would like to address each of your concerns in detail:
>
> > W1
>
> Thank you for raising this important issue about the framework's generalizability. We would like to clarify several aspects:
> 1. Design Philosophy:
> - Our proxy-centric alignment is inspired by human interaction patterns in knowledge-intensive tasks, where information gathering and reasoning are fundamental operations.
> - The core design focuses on how proxy can align retrievers and LLMs through planning and reasoning to collect information, rather than being strictly QA-specific.
>
> 2. Preservation of LLM Capabilities:
> - Importantly, our framework does not fine-tune the LLM, preserving its general capabilities (e.g., writing, summarization).
> - The agents serve as information gathering and coordination proxies, which are inherently applicable to various knowledge-intensive tasks beyond QA.
>
> We will include this as an important direction for future work, while maintaining that the current design principles are fundamentally task-agnostic.
>
> > W2
>
> Thank you for raising this important issue. We would like to address your concerns from multiple aspects:
> 1. **Training Efficiency:**
> - Our approach **does not introduce significant additional training overhead** compared with standard PPO.
> - Traditional RL methods typically require **sampling multiple independent trajectories per question in parallel**, and our approach may reuse partial trajectories during rollout. In SGLang inference system, the reused context can be efficiently cached for further inference. Algorithmically, our approach just **redistributes this sampling effort from the question level to the action level**, maintaining a similar computational budget.
>
> 2. **Inference Efficiency:**
> - `Figure 3` shows our method does not introduce substantial inference latency.
> - This efficiency is achieved through our Decision Maker, which dynamically allocates optimal strategies to balance computation and performance.
> - `Figure 5` shows how different strategies evolve during RL iterations, providing transparency into our method's adaptation.
>
> 3. **Regarding Baseline Comparisons:**
> - While works [1,2] are not yet open-sourced for faithful reproduction, we have included another reranking work in `Tables 2/3`.
> - We chose QueryRewriting as our efficiency baseline due to its parameter efficiency (1.5B) and consistent stability across scenarios.
>
> We appreciate these suggestions and will incorporate the related works [1,2] to better position our work.
>
> > Q1
>
> Thank you for this detailed question about computational efficiency. Let us break down the number of forward passes required for each strategy:
> 1. **Empirical Evidence:**
> - `Figure 6` provides `detailed distributions of inference depths` across different datasets
> - `Figure 3` shows the inference latency of C-3PO compared to baselines
>
> 2. **Forward Passes by Strategy:**
> - [No Retrieval]: 1 proxy pass + 1 LLM pass
> - [Retrieval]: 2 proxy passes + 1 LLM pass
> - [Planning]: 2 LLM passes + variable proxy passes (distribution shown in Figure 6)
> - Note that Proxy are lightweight (0.5/1.5B) and LLM can be 7/72B
>
> This strategic allocation of computational resources allows us to maintain efficiency and achieve superior performance. The actual number of forward passes is optimized for each specific query rather than using a fixed number for all cases.
>
> > Q2
>
> Thank you for this insightful observation. The [Planning] strategy, while powerful, involves collecting additional information that may introduce more noise and potentially mislead the LLM. Meanwhile, for many RAG datasets, a well-crafted query combined with effective filtering ([Retrieval] in our C-3PO) might suffice, especially when search engines can retrieve the necessary information in a single pass. This suggests that the optimal strategy depends on the alignment between dataset and proxy capabilities rather than following a one-size-fits-all approach.
>
> > Q3
>
> Thank you for this thoughtful question about extending our framework beyond QA tasks. While our C-3PO framework is specifically designed for **knowledge-intensive tasks** where proxy-centric alignment between retrieval and LLM components is crucial, the tasks in [3] primarily focus on **logical reasoning** that may not heavily rely on external knowledge. For such pure logical reasoning tasks, our retrieval-oriented multi-agent system might offer limited benefits in its current form.
>
> However, we believe our framework could be adapted for logical reasoning tasks by:
> - Combining a reasoning verification agent
> - Integrating training approaches similar to Deepseek-R1 for pure reasoning tasks
>
> We appreciate this suggestion as it opens up interesting directions for future research.
>
> We sincerely appreciate your detailed review and thoughtful questions. We believe addressing these points has helped strengthen our paper. We look forward to your further feedback.

---

### Official Review · Reviewer_TgQU · 2025-03-13

**Overall Recommendation:** 3

**Summary:**

The paper proposes C-3PO, a plug-and-play multi-agent system used to enhance the alignment of retrievers and LLMs in RAG systems. Specifically, C-3PO consists of three LLM agents: a reasoning router designed to determine the reasoning strategy for a specific question, an information filter agent used to identify relevant documents from retrieved ones, and a decision maker agent designed to determine the optimal action based on the current state. To optimise these agents, the paper uses reinforcement learning to train these agents and proposes a simple tree-structured rollout approach for robust on-policy learning. For the tree-structured rollout, it computes reward by enumerating all possible reasoning strategies for each question. Experimental results on both in-domain and out-of-domain datasets validate the effectiveness of the proposed  C-3PO.

**Claims And Evidence:**

The claims are well-supported by the experimental results.

**Essential References Not Discussed:**

The following iterative/adaptive RAG models are missing from the paper:
1. Trivedi, Harsh, et al. "Interleaving retrieval with chain-of-thought reasoning for knowledge-intensive multi-step questions." ACL 2023.
2. Jiang, Zhengbao, et al. "Active retrieval augmented generation." EMNLP 2023.
3. Su, Weihang, et al. "DRAGIN: Dynamic Retrieval Augmented Generation based on the Information Needs of Large Language Models." ACL 2024.
4. Jeong, Soyeong, et al. "Adaptive-rag: Learning to adapt retrieval-augmented large language models through question complexity." NAACL 2024.

**Experimental Designs Or Analyses:**

The experimental design appears reasonable and well-structured.

**Methods And Evaluation Criteria:**

The proposed method is sold. However, the paper relies on an LLM (see Appendix D.2) to evaluate the generated answers, which raises concerns about potential biases and reliability. It is unclear which LLM is used for evaluation and how different LLMs would affect the results.  The paper should also provide results on existing QA evaluation metrics, such as Exact Match (EM) and F1-score, to offer a more standardized and quantitative assessment of the answers.

**Other Comments Or Suggestions:**

No.

**Other Strengths And Weaknesses:**

Strengths:
1. The paper is well-written and easy to follow.
2. The proposed C-3PO seems novel. Experimental results on six in-domain datasets and two out-of-the domain datasets validate the effectiveness of the proposed C-3PO.
3. Ablation studies are conducted to verify the effectiveness of each component.

Weaknesses:
1. The proposed tree-structured rollout method incurs high computational cost, as it requires exploring all possible reasoning trajectories for each question. This exhaustive search significantly increases the training overhead, limiting its practicality.

2. The paper states that it employs a warm-up phase to train the multi-agent system. Despite some descriptions of the supervised warm-up phase, the details remain unclear. Although the appendix provides some additional information, it does not fully explain the specifics of the training process, including the training data and training methodology.

3. The introduction of the evaluation metrics should be moved from Appendix to the main paper.

4. A major concern is the use of LLM for evaluation, raising questions about bias and reliability. It is unclear why conventional QA metrics such as Exact Match and F1 are not reported.

**Questions For Authors:**

Please see the questions in above sections.

**Relation To Broader Scientific Literature:**

Existing works on leveraging intermediate component to bridge the gap between retrievers and LLMs focus on optimising a single task in isolation, which may lead to suboptimal performance. The paper proposes C-3PO to faciliate seamless communication between retrievers and LLMs.

**Theoretical Claims:**

There is no theoretical analysis in the paper.

---

> ### Author Rebuttal · Authors · 2025-03-31
>
> Dear Reviewer TgQU,
>
> Thank you for your thorough and constructive review of our paper. We would like to address each of your concerns in detail:
>
> > W1
>
> We appreciate your concern about computational efficiency. We would like to clarify that our tree-structured rollout **does not introduce additional computational overhead** compared to standard RL:
>
> - Traditional RL methods typically require **sampling multiple independent trajectories per question in parallel**, and our approach may reuse partial trajectories during rollout. In SGLang inference system, the reused context can be efficiently cached for further inference. Algorithmically, our approach just **redistributes this sampling effort from the question level to the action level**, maintaining a similar computational budget.
>
> - The tree structure actually provides **several advantages**:
>   - It enables more systematic exploration of the action space
>   - It allows for expectation-based credit assignment
>   - It reduces the variance in training compared to random sampling
>
> - We can also control the breadth and depth of the tree to balance between exploration and computational cost (Eq. 4), making it flexible for different computational budgets.
>
> Therefore, while our approach may appear computationally intensive at first glance, it actually offers a more structured and efficient way to explore the action space within the same computational constraints as traditional RL methods.
>
> > W2
>
> We apologize for any confusion regarding the warm-up phase. We would like to clarify several key points:
> - As mentioned in `Section 5.2 and Appendix A.2`, we collect seed data through rejection sampling from Qwen2-72b-instruct, specifically gathering 2 correct solutions for each question.
>
> - The detailed training hyper-parameters are provided in `Table 4`.
>
> - To further validate the effectiveness, we conducted comparative experiments between C-3PO-RL and C-3PO-ICL in `Table 8`. These results demonstrate the feasibility of our warm-up strategy.
>
> We hope these clarifications address your concerns about the warm-up phase implementation.
>
> > W3
>
> We agree that the evaluation metrics deserve more prominence in the main text. We will move a concise version of the evaluation metrics `from Appendix D.2 to Section 6.1`, making these important details more accessible while maintaining the paper's flow and readability.
>
> > W4
>
> Thank you for raising this important point about evaluation methodology. We would like to address this concern from multiple aspects:
>
> 1. **Additional EM Results**:
>
> We have conducted additional experiments using the EM metric. The results show that on the EM metric, C-3PO still achieves significant improvements over all baselines:
>
> | Methods|2Wiki|HQA|Musique|NQ|PopQA|TQA|AVG|
> |-|:-:|:-:|:-:|:-:|:-:|:-:|:-:|
> |Direct|36.4|36.8|17.5|44.1|25.1|73.4|38.88|
> |Standard|26.1|41.3|31|52.1|38.1|73.8|43.73|
> |REPLUG|25.2|39.8|24|43.2|37.7|74.3|40.7|
> |Self-RAG| - | - | - |41.7|40.5|74.9|52.36|
> |InstructRAG|45.9| - | - |51.6|40.9|75.6|53.5|
> |Auto-RAG|44.7|41.3| - |43.8|39.2|72.1|48.22|
> |ReRanker|29.8|37.6|19.4|47.6|20.7|73.3|38.06|
> |QueryRewrite|42.9|47.3|44.5|60.6|40.3|79.1|52.45|
> |SKR-KNN|38.6|54.8|37.7|56.2|38.6|73.5|49.9|
> |SlimPLM| - | - |19.8|57.6| - |76.4|51.26|
> |C-3PO-0.5B|60.5|61.1|50.1|65.9|52.7|80.3|61.76|
> |C-3PO-1.5B|63.7|63|54.8|67.7|53.8|82|64.16|
>
> 2. **Limitations of Traditional Rule Based Metrics**:
> - Through our preliminary studies, we found that rule based metrics, such as EM, can be unreliable, especially when working with frozen LLMs that may express correct answers in unpredictable formats.
> - These inaccurate rewards from strict rule based matching could potentially harm the training of reinforcement learning.
>
> 3. **Adoption of LLM-based Evaluation:**
> - Recent famous benchmarks like FreshQA and Humanity Last Exam (HLE) increasingly adopt LLM-based evaluation to capture semantic correctness beyond exact matching.
> - This trend reflects the community's recognition of the limitations of traditional metrics for complex QA tasks.
>
> 4. **Reliability of Our Evaluation**:
>
> We conducted rigorous human verification of `Qwen2-72B-instruct`'s evaluation capabilities.
>
> We understand the importance of using standardized metrics. However, we believe that combining both traditional and LLM-based evaluation provides a more comprehensive assessment of model performance. We appreciate this feedback and have enhanced our evaluation section accordingly.
>
> > References Not Discussed
>
> We sincerely thank you for suggesting these valuable references. We will incorporate these citations and related discussions in our revised manuscript to better position our work in RAG.
>
> We appreciate again for your time and effort in reviewing our paper. We believe that addressing these concerns has helped strengthen our work, and we hope our responses have satisfactorily addressed your questions. We look forward to your further feedback.

---

### Official Review · Reviewer_B7TK · 2025-03-14

**Overall Recommendation:** 2

**Summary:**

The paper proposes C-3PO,  which introduces a multi-agent system that optimizes retrieval, query generation, and information filtering. It uses multi-agent reinforcement learning (MARL) with tree-structured rollout and Monte Carlo credit assignment. Experiments show that C-3PO significantly enhances RAG performance across in-domain and out-of-distribution datasets, demonstrating its plug-and-play flexibility and strong generalization capabilities​

**Claims And Evidence:**

Most of the claims in this paper are supported by the evidence.

Some issues:
- The paper does not compare against certain related baselines that also use tree-based rollout [3] or multi-agent training for RAG [1, 2], making it unclear how C-3PO improves over existing methods.
- There is no detailed analysis of the role of each agent in the system. While Table 3 may provide some insights, the experimental setup is unclear, and it is not explicitly explained what each row represents.
- The performance gain of tree-structured rollout over standard reinforcement learning appears marginal in Figure 2, raising concerns that the proposed approach may be overly complex without substantial benefits.


References:

[1] Chen, Yiqun, et al. "Improving Retrieval-Augmented Generation through Multi-Agent Reinforcement Learning." arXiv preprint arXiv:2501.15228 (2025).

[2] Shao, Zhihong, et al. "Enhancing retrieval-augmented large language models with iterative retrieval-generation synergy." arXiv preprint arXiv:2305.15294 (2023).

[3] Jiang, Jinhao, et al. "RAG-Star: Enhancing Deliberative Reasoning with Retrieval Augmented Verification and Refinement." arXiv preprint arXiv:2412.12881 (2024).

**Essential References Not Discussed:**

RAG with multi-agent systems:

Chen, Yiqun, et al. "Improving Retrieval-Augmented Generation through Multi-Agent Reinforcement Learning." arXiv preprint arXiv:2501.15228 (2025).

Shao, Zhihong, et al. "Enhancing retrieval-augmented large language models with iterative retrieval-generation synergy." arXiv preprint arXiv:2305.15294 (2023).

Zhu, Junda, et al. "ATM: Adversarial Tuning Multi-agent System Makes a Robust Retrieval-Augmented Generator." arXiv preprint arXiv:2405.18111 (2024).

**Experimental Designs Or Analyses:**

See above sections for details.

**Methods And Evaluation Criteria:**

The choice of datasets is reasonable.

However, it is unclear why EM/F1/Accuracy scores were not used as the final performance metrics, given that they are widely adopted in prior work (numerous references support this). It is recommended to at least provide some numbers on one/more of these metrics.

**Other Comments Or Suggestions:**

N/A

**Other Strengths And Weaknesses:**

Strengths:

- Clear Modular Design for Multi-Agent Collaboration

- Strong Performance on RAG Tasks

- Detailed prompt format, simple,mentation details provided

Weaknesses:

- Additional studies using alternative metrics (e.g., EM/F1) and inference efficiency analysis would strengthen the empirical results.

- The method should be tested on a wider range of LLM APIs and local models to assess its generalizability across different deployment settings.

- Including additional baselines that use tree-based rollout or multi-agent training would provide a more comprehensive comparison.

**Questions For Authors:**

See above comments.

**Relation To Broader Scientific Literature:**

This paper proposes an online RL training method that is reasonable and has some novelty. However, the added complexity raises concerns about whether the performance gains justify the additional computational cost.

**Theoretical Claims:**

n/a

---

> ### Author Rebuttal · Authors · 2025-03-31
>
> Dear Reviewer B7TK
>
> We sincerely appreciate your thorough review. We have carefully addressed each of your concerns below:
> > Issue 1 & W3
>
> We appreciate the mentioned related works, and would like to clarify several important points:
> 1. First, we acknowledge the importance of these works. We will **cite these related works and incorporate detailed discussions in our revised version**.
> 2. Regarding the timeline and reproducibility of mentioned related works:
> - [1] is published on **Jan 25, 2025**, 2 days later than ICML abstract submission deadline **Jan 23, 2025**.
> - [3] is published on Dec 17, 2024, which can be considered **concurrent work**.
> - For [2] and [3], despite their relevance, **the absence of publicly available implementation** makes faithful reproduction challenging.
> 3. Our evaluation have covered 3 major baseline categories (retriever/LLM fine-tuning and intermediate approaches) across 6 in-domain and 2 OOD datasets, demonstrating thorough effectiveness and generalization.
>
> > Issue 2
>
> We apologize for any confusion, while the experimental setup of each row in Table 3 was presented in `Line 418-426`, we would like to provide further clarification:
> - [No Retrieval]: Relies solely on LLM's inherent knowledge
> - [Retrieval]: Employs single retrieval-filter loop
> - [Planning]: Utilizes multi-step reasoning
>
> The full C-3PO system's ability to adaptively select strategies leads to robust performance across different datasets.
>
> > Issue 3
>
> We appreciate the reviewer's careful examination of our tree-structured rollout. We would like to provide additional clarification:
> 1. Regarding the performance gains:
> - We observe substantial gains on challenging tasks such as Musique, HQA, and PopQA.
> - While our method **enhances agent decision-making instead of directly answering the question**, the performance ceiling **ultimately** depends on the LLM (remains frozen in C-3PO). Our approach still outperforms many recent methods that fine-tune LLMs (AutoRAG) as shown in Tables 1/2.
>
> 2. On complexity concerns:
> - We would like to emphasize that our tree-structured rollout **does not introduce additional computational overhead** compared to standard RL.
> - Traditional RL methods typically require **multiple independent sampling trajectories per question in parallel**, and our approach may reuse partial trajectories during rollout. In SGLang inference system, reused context can be efficiently cached. Algorithmically, our approach **redistributes these sampling efforts from the question level to the action level**, enabling more efficient credit assignment in multi-agent systems through expectation-based reward distribution.
> - We can also control the breadth/depth of the tree to balance exploration and cost (Eq. 4), making it flexible for various computational budgets.
>
> We believe the clarifications show that our approach offers meaningful improvements and maintains computational efficiency.
>
> > Eval Criteria
>
> We appreciate the reviewer's suggestion regarding evaluation metrics. We would like to clarify our choice of metrics and provide additional results:
>
> 1. **Limitations of EM metrics**:
> - Through our preliminary studies, we observed that rule based metrics, such as EM, can be unreliable, especially when using frozen LLMs that may express correct answers in varied formats.
> - Inaccurate rewards from strict rule based matching could potentially harm the RL training.
>
> 2. **The choice of LLM-based evaluation**:
> - Recent benchmarks, such as FreshQA, HLE (Humanity's Last Exam), have increasingly adopted LLM-based evaluation due to its ability to capture semantic correctness beyond EM.
> - In our human verification process, we found that Qwen2-72B-instruct demonstrates high accuracy in assessment, making it as a more reliable source for both evaluation and RL rewards.
>
> 3. **Additional EM Results**:
> - To address this concern, we provide some EM results (due to limited chars, a full table can refer to the response of W4 for Reviewer TgQU).
>
> |Methods|2Wiki|HQA|Musique|NQ|PopQA|TQA|AVG|
> |-|:-:|:-:|:-:|:-:|:-:|:-:|:-:|
> |Standard|26.1|41.3|31|52.1|38.1|73.8|43.73|
> |Auto-RAG|44.7| 41.3| - |43.8|39.2|72.1|48.22|
> |C-3PO-0.5B|60.5|61.1|50.1|65.9|52.7|80.3|61.76|
> |C-3PO-1.5B|63.7|63|54.8|67.7|53.8|82|64.16|
>
> > W1
>
> Regarding inference efficiency, we already presented a detailed analysis in `Section 6.5 and Figure 3`. It shows that C-3PO achieves the best performance-efficiency trade-off.
>
> > W2
>
> We appreciate the suggestion and would like to clarify that our evaluation **already covers a diverse range of LLMs across different scales and types**, such as `Qwen2-7B`, `Qwen2-72B`, `Llama3.3-70B`, and `GPT4o-mini` (commercial API), as shown in Tables 1/2.
> While we acknowledge that testing on other commercial APIs like Claude and o1 would be interesting, the significant costs make such extensive evaluation prohibitively expensive.
>
> We sincerely thank you for your detailed comments and hope our responses have adequately addressed your concerns.

---

### Decision · Program_Chairs · 2025-05-01

**Decision:**

Accept (poster)

**Comment:**

The paper proposes C-3PO, a proxy-centric framework that facilitates communication between retrievers and LLMs through a lightweight multi-agent system. The reviewers found the method to be interesting and reasonable, yielding strong empirical performance. Some reviewers raised concerns including discussion with similar (and concurrent) works, evaluation metrics, and LLM-based evaluation, which were addressed by the authors quite comprehensively in the rebuttal.

Minor note: Some references used the arXiv version instead of the conference proceeding version (e.g., RankRAG is a NeurIPS'24 paper and InstructRAG is an ICLR'25 paper), which are better to be updated.